# CAR affinity modulates the sensitivity of CAR-T cells to PD-1/PD-L1-mediated inhibition

Irene Andreu-Saumell [1,5], Alba Rodriguez-Garcia [1,5] ✉, Vanessa Mühlgrabner[2], Marta Gimenez-Alejandre[1], Berta Marzal[1], Joan Castellsagué[1], Fara Brasó-Maristany [1], Hugo Calderon [1], Laura Angelats[1,3], Salut Colell [1], Mara Nuding[1], Marta Soria-Castellano [1], Paula Barbao [1], Aleix Prat [1,3,4], Alvaro Urbano-Ispizua[1,3], Johannes B. Huppa [2] & Sonia Guedan [1] ✉

Chimeric antigen receptor (CAR)-T cell therapy for solid tumors faces significant hurdles, including T-cell inhibition mediated by the PD-1/PD-L1 axis. The effects of disrupting this pathway on T-cells are being actively explored and controversial outcomes have been reported. Here, we hypothesize that CAR-antigen affinity may be a key factor modulating T-cell susceptibility towards the PD-1/PD-L1 axis. We systematically interrogate CAR-T cells targeting HER2 with either low (LA) or high affinity (HA) in various preclinical models. Our results reveal an increased sensitivity of LA CAR-T cells to PD-L1-mediated inhibition when compared to their HA counterparts by using in vitro models of tumor cell lines and supported lipid bilayers modified to display varying PD-L1 densities. CRISPR/Cas9-mediated knockout (KO) of PD-1 enhances LA CAR-T cell cytokine secretion and polyfunctionality in vitro and antitumor effect in vivo and results in the downregulation of gene signatures related to T-cell exhaustion. By contrast, HA CAR-T cell features remain unaffected following PD-1 KO. This behavior holds true for CD28 and ICOS but not 4-1BB co-stimulated CAR-T cells, which are less sensitive to PD-L1 inhibition albeit targeting the antigen with LA. Our findings may inform CAR-T therapies involving disruption of PD-1/PD-L1 pathway tailored in particular for effective treatment of solid tumors.

Clinical outcomes achieved until date with CAR-T cell therapy for the treatment of solid tumors are yet far from the unprecedented success witnessed in hematologic malignancies[1]. In spite of this, recent works provide clear evidence of objective antitumor responses in patients with hard to treat solid tumors[2–4]. These results are highly encouraging and provide proof of the potential of CAR-T cells in this setting. Nevertheless, several obstacles remain to be addressed, including T cell inhibition within the suppressive tumor microenvironment (TME)[5].

One of the most prominent and well-studied T cell inhibitory axis is the PD-1/PD-L1 immune checkpoint pathway. T cell activation

[1]Oncology and Hematology Department, Fundació Clínic Recerca Biomédica- IDIBAPS, Barcelona, Spain. [2]Medical University of Vienna, Center for Pathophysiology, Infectiology and Immunology, Institute for Hygiene and Applied Immunology, Vienna, Austria. [3]Department of Medicine, University of Barcelona, Barcelona, Spain. [4]Institute of Cancer and Blood Diseases, Hospital Clínic de Barcelona, Barcelona, Spain. [5]These authors contributed equally: Irene Andreu-Saumell, Alba Rodriguez-Garcia. ✉e-mail: rodriguez6@recerca.clinic.cat; sguedan@recerca.clinic.cat

following antigen recognition results in PD-1 upregulation, along with an intracellular signaling cascade that leads to the release of Th1 cytokines. These cytokines, in turn, induce the upregulation of the inhibitory ligand PD-L1 on tumor cells but also on other cell populations within the TME. The interaction between PD-1 on T cells and PD-L1 on tumors ultimately leads to T cell suppression[6,7]. As these activated T cells are potentially tumor-specific infiltrating T cells (either endogenous or adoptively transferred T cells modified to express tumor-specific TCR or CARs), preventing the binding between PD-1 and PD-L1 might rescue antitumor T cell cytotoxicity and result in increased efficacy of cell-based immunotherapies.

A variety of methodologies including immune checkpoint blockade antibodies (in combination or secreted by the CAR-T cells themselves), downregulation of PD-1 (by shRNA or by relocating PD-1 to Golgi/ endoplasmic reticulum (ER) using retention peptides), genetic disruption (by TALEN or CRISPR/Cas9) or dominant negative receptors (DNR) have been used to increase the potency of CAR-T cells. Although the majority of reports demonstrate an advantage of targeting the PD-1/PL1 axis in terms of increased anti-tumor properties[8-17], it is worth noting that some studies have raised concerns about adverse effects associated with long-term PD-1 disruption, including induction of T cell exhaustion and impaired persistence[18,19], implying certain discrepancies in the field. Conflicting findings observed in these investigations can be attributed to several factors, including variability in (i) PD-1 disruption methodology, (ii) preclinical models and (iii) CAR constructs employed.

In this work, we hypothesize that CAR affinity is a key underexplored factor modulating T cell sensitivity to PD-1/PD-L1 axis. To address the model variability issue and gain deeper understanding on how CAR affinity for the targeted antigen might influence this pathway, we develop a preclinical model of tumor cell lines engineered to express different PD-L1 densities (absent, low, or high) for systematic interrogation of different CAR configurations both in vitro and in vivo. We also develop a synthetic model of glass-supported lipid bilayers (SLBs) with controlled amounts of both target antigen and PD-L1 molecules. Using these preclinical models, we explore the effects of inhibiting the PD-1/PD-L1 axis on CAR-T cells targeting their cognate antigen with either low (LA) or high affinity (HA) and comprising different co-stimulatory domains (CD28, ICOS, and 4-1BB). We find that LA CAR-T cells are more sensitive to PD-1/PD-L1 axis-mediated inhibition compared to HA CARs. Consequently, PD-1 disruption enhances the functionality of LA CAR-T cells, while it does not provide an advantage to HA CAR-T cells. This is true for CD28 and ICOS costimulation domains, while 4-1BB co-stimulated CAR-T cells are intrinsically more resistant to PD-L1-mediated inhibition regardless of the affinity for the targeted antigen.

## Results

### PD-1/PD-L1 inhibition restores in vitro functionality of LA but does not impact HA CAR-T cells

To study the role of PD-1/PD-L1 axis on CAR-T cells under controlled conditions, we first generated a tumor model based on the ovarian cancer cell line SKOV3, which was engineered to express varying PD-L1 densities (negative, low or high). We validated HER2 expression across all generated cell lines (Supplementary Fig. 1a) and confirmed expected patterns of PD-L1 expression both in vitro (Fig. 1a) and in xenograft tumors in vivo (Fig. 1b). Moreover, we compared them to those found in wild-type (WT) tumor cell lines from different tissues, including SKOV3, either at basal levels or after co-culture with CAR-T cells (Supplementary Fig. 1b, c). We validated that our model is representative of the various densities of PD-L1 found physiologically in different tumor cell lines. In parallel, we generated CD28-costimulated CAR-T cells targeting HER2 either with high affinity (HA) by using the trastuzumab-based 4D5 scFv, or with low affinity (LA) by using a

mutated version of the previous with a ~2000-fold reduced affinity, named 4D5.5 (Fig. 1c)[20].

We assessed the expression levels of surface CAR (Fig. 1d and Supplementary Fig. 1d) and PD-1 (Fig. 1e). Both LA and HA mock CAR-T cells exhibited comparable levels of CAR and PD-1 expression of approximately 70%. This substantial PD-1 expression was indicative of a robust T-cell activation during primary expansion. Subsequently, by employing CRISPR/Cas9-mediated gene editing to KO PD-1, we observed a significant reduction in PD-1 expression levels (Fig. 1e). We consistently achieved ablation efficiencies of approximately 80% in all normal donors used (Fig. 1f). PD-1 deletion did not impact T-cell expansion (Supplementary Fig. 1f) or CAR-mediated activation, as evidenced by similar population doublings and uniform CD25 upregulation in both edited and mock CAR-T cells following restimulation (Supplementary Fig. 1g).

To determine the effects of PD-1 ablation on CAR-T cell function, we co-cultured mock or PD-1 KO HER2-28Z CAR-T cells of LA or HA with our PD-L1 cellular model and measured cytokine secretion. In the absence of PD-L1 expression by tumor cells, both mock and PD-1 KO LA CAR-T cells released similar amounts of IFN-γ. Conversely, PD-L1 expressed by SKOV3 either at low or high levels suppressed IFN-γ secretion by mock CAR-T cells. This loss in cytokine release was restored by genetic disruption of PD-1 (Fig. 1g). In contrast, PD-1 KO in HA CAR-T cells led to non-significant increases in cytokine release, implying a higher resistance of HA CAR-T cells to PD-L1 mediated inhibition (Fig. 1h). Similar results were observed when the PD-1/PD-L1 axis was targeted in LA and HA CAR-T cells using blocking antibodies against PD-1 or PD-L1 (Fig. 1i, j) and for the secretion of IL-2 (Supplementary Fig. 2a–d). We validated these findings in an alternative pair of CARs targeting FRβ with different affinities (Fig. 1c and Supplementary Fig. 3a–c)[21]. By using the SKOV3-based PD-L1 cellular model in which we overexpressed FRβ (Supplementary Fig. 3d), we found significantly higher levels of IFN-γ secreted by PD-1 KO LA CAR-T cells as compared to mock in the presence of PD-L1 (Fig. 1k, left panel). These differences were not observed in co-culture with the PD-L1 KO cell line or between PD-1 KO and mock HA CAR-T cells, which released similar levels of IFN-γ regardless of PD-L1 presence (Fig. 1k, right panel). While the differences in affinity were less pronounced for the FRβ-targeting CAR pair (21.89-fold), these findings align with our observations from the HER2 model.

We next tested the hypothesis that lack of PD-1 expression promotes enhanced proliferation in CAR-T cells. To this end, we co-cultured mock or PD-1 KO LA and HA HER2-28Z CAR-T cells with the breast cancer cell line HCC1954, selected due to its high expression levels of HER2 and PD-L1 (Supplementary Fig. 1c, d). Our findings revealed a two-fold increase in the proliferation of LA PD-1 KO CAR-T cells as compared to mock CAR-T cells 6 days after stimulation. In line with the cytokine secretion results, PD-1 KO did not provide a proliferative advantage to HA CAR-T cells (Fig. 1l).

### PD-1 KO increases efficacy of LA but not HA HER2-28Z CAR-T cells in vivo

We next aimed to evaluate the impact of PD-1/PD-L1 axis on the therapeutic potential of HER2-28Z CAR-T cells in vivo. NSG mice bearing SKOV3 subcutaneous tumors expressing different levels of PD-L1 were treated with a single dose of control T cells, mock or PD-1 KO CAR-T cells. In line with cytokine release data obtained in vitro, both mock and PD-1 KO LA CAR-T cells showed similar anti-tumor activity and efficiently eliminated tumors that did not express PD-L1 (Fig. 2a, left panel). However, even low levels of PD-L1 expression impaired the anti-tumor efficacy of LA HER2 CAR-T cells. PD-1 ablation significantly enhanced anti-tumor responses, resulting in complete regressions in 90% of tumors expressing either low or high levels of PD-L1 (Fig. 2a, middle and right panels). We validated these results with CAR-T cells in combination with anti-PD-L1 blocking antibodies in high PD-L1-

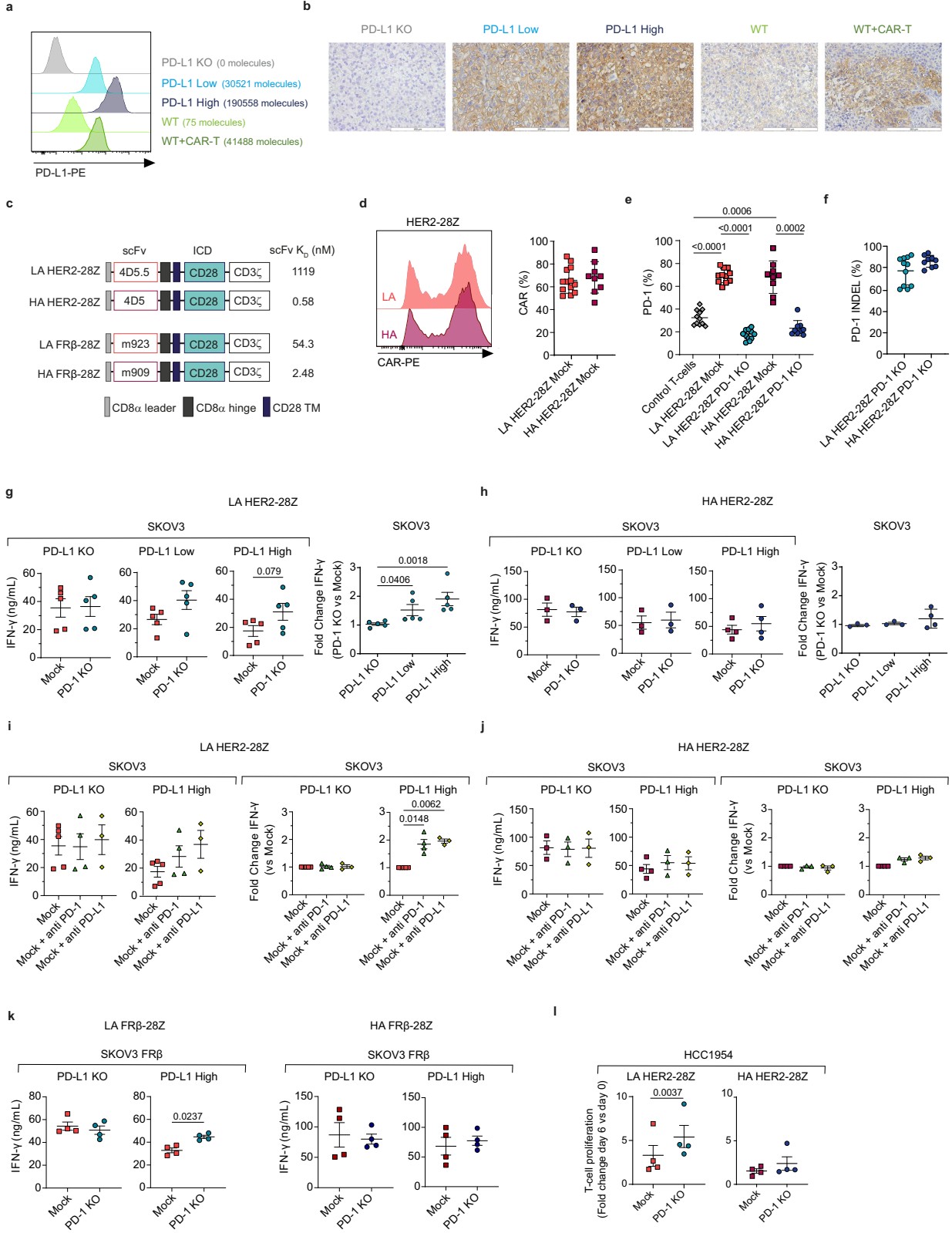

expressing SKOV3 cells. Even though the combination with antibodies improved the anti-tumor effect of CAR-T cells alone, PD-1 KO CAR-T cells still showed the best efficacy (Fig. 2b, c). Regarding HA CAR-T cells, both mock and PD-1 KO were able to eliminate nearly all tumors, including those with the highest PD-L1 expression levels (Fig. 2d). We then measured the anti-tumor efficacy in WT SKOV3 cells, expressing physiological levels of PD-L1. Again, PD-1 KO provided a significant

advantage only to LA CAR-T cells (Fig. 2e), while the efficacy of HA CAR-T cells was not further improved (Fig. 2f). As HA mock CAR-T cells were able to eliminate tumors in all models tested, we repeated the in vivo experiment in SKOV3 WT cell line using more challenging conditions and treating larger tumors. We consistently found no significant differences between mock and PD-1 KO CAR-T cells, except for a noticeable reduction in overall efficacy observed in both groups

**Fig. 1 | PD-1 KO restores LA HER2-28Z CAR-T cell function in vitro but does not affect HA HER2-28Z CAR-Ts. a** Expression and quantification of PD-L1 molecules in SKOV3 cells expressing variable PD-L1 densities and compared to wild type (WT) cells alone or co-cultured with CAR-T cells for 48 h as assessed by flow cytometry. **b** Immunohistochemical staining of PD-L1 in SKOV3 PD-L1 KO, Low, High, and WT tumors treated with control T-cells or CAR-T cells at day 20–70 post-implantation in mice. Representative images from $n = 2$ tumors per group are shown. Scale bar, 200 μm. **c** Schematic overview of CAR constructs used with their corresponding affinity values. HER2-28Z CAR-T cells with LA or HA and with or without PD-1 genome editing were generated from 9 to 13 healthy donors. CAR (**d**) and PD-1 (**e**) expression were quantified by flow cytometry and the efficiency of PD-1 knock-out (**f**) was quantified by using ICE tool (Synthego). LA (**g**, **i**) or HA (**h**, **j**) HER2-28Z CAR-T cells were co-cultured with indicated SKOV3 cells for 24 h. The PD-1/PD-L1 axis

was inhibited by knocking out PD-1 (**g**, **h**) or by addition of blocking antibodies (**i**, **j**). IFN-γ production was analyzed by ELISA ($n = 3$–$5$). Data is represented as absolute levels (left panel) or as fold change versus mock (right panel). **k** LA (left panel) or HA (right panel) FRβ−28Z CAR-T cells were co-cultured with SKOV3 expressing FRβ and indicated PD-L1 densities. IFN-γ production was quantified by ELISA ($n = 4$). **l** T-cell proliferation of LA (left panel) or HA (right panel) HER2-28Z CAR-T cells following co-culture with HCC1954. Fold change of absolute T-cell numbers at day 6 versus day 0 is represented ($n = 4$). Data in (**d**–**l**) are pooled from independent experiments where each dot represents CAR-T cells generated from a different donor (n) and represented as mean ± SD (in **d**–**f**) or mean ± SEM (in **g**–**l**). $p$ values by a two-tailed paired $T$ test (**e**, **g**, **h** for absolute levels graphs, **k**, **l**), one-way ANOVA (**g**, **h** for fold change graphs) or two-tailed one sample $T$ test (**i**) are indicated. Source data and exact n values for each group in (**d**–**j**) are provided as a Source Data file.

(Supplementary Fig. 4). Similar results in terms of enhanced anti-tumor activity by PD-1 KO LA HER2-28Z CAR-T cells were observed in the breast cancer model HCC1954 (Fig. 2g). Collectively, these findings indicate that genetic disruption of PD-1 enhances the anti-tumor effect of LA HER2-28Z CAR-T cells in tumors expressing PD-L1, while it does not impact the efficacy of HA CAR-T cells.

## Affinity-based differences to inhibition by PD-L1 are maintained in a lipid bilayer system

To ensure that differences observed between LA and HA CAR-T cells upon PD-1 ablation were not due to intrinsic changes in tumor cells derived from differential expression of PD-L1, we utilized a supported lipid bilayer system (SLB) functionalized with the intercellular adhesion molecule 1 (ICAM-1) to facilitate cell attachment, HER2 for CAR recognition and PD-L1 at titrated densities (Fig. 3a)[22,23]. Presence of PD-1 on the CAR-T cell membrane was confirmed by flow cytometry before adding the cells to the system, ensuring its interaction with the PD-L1 in the bilayer (Supplementary Fig. 5). After exposing LA and HA mock or PD-1 KO CAR-T cells to the SLBs, we determined their respective activation levels via IFN-γ secretion. In line with results obtained in the cellular model, IFN-γ release by LA mock CAR-T cells was reduced in the presence of high densities of PD-L1, and this inhibition was reversed by PD-1 KO (Fig. 3b, left panel and 3c). In contrast, secretion of IFN-γ by both mock and PD-1 KO HA CAR-T cells remained unaltered at all PD-L1 concentrations tested (Fig. 3b, right panel and 3d). Overall, differences in IFN-γ secretion between LA and HA HER2-28Z CAR-T cells in the presence of PD-L1 persisted in this system, thereby confirming that they were indeed attributable to PD-L1.

## PD-1 KO induces deeper changes in the transcriptome of LA as compared to HA HER2-28Z CAR-T cells

In order to characterize the molecular mechanisms behind the different effects of PD-1 disruption in LA and HA HER2-28Z CAR-T cells, we compared the transcriptomic profile of mock and PD-1 KO CAR-T cells following antigen recognition by using the nCounter® CAR-T Characterization Gene Expression Panel (Nanostring Technologies). Knocking PD-1 out of LA HER2-28Z CAR-T cells resulted in statistically significant downregulation of 20 genes and upregulation of 13 genes out of the 780 genes analyzed in the panel (Fig. 4a, b, left panels). Within upregulated genes in PD-1 KO versus mock LA HER2 CAR-T cells we found FosB, a transcription factor that has been previously reported to be decreased in exhausted T cells during chronic viral infection[24] while increased in CAR-T cells from responding as compared to non-responding patients[25]. Other upregulated genes included T-cell activation-related genes such as effector cytokines (IFN-γ, TNF, IL-2, CSF2/GM-CSF or CLCF1), chemokines (CCL3, CCL4, CCL20, XCL1/2) or co-stimulatory molecules (TNFSF9/4-1BBL). Genes that were down-regulated in the PD-1 KO CAR-T cells included the transcription factor MAF, regulon driver of T cell exhaustion[24,26], genes related to type I and II IFN signaling (IRF9, ADAR, SP100, SOCS2, ISG15, STAT1, STAT2, IFIT1, IRF7, PML, IFI35), which have been recently linked to CAR-T cell

dysfunction[27–29], CD68 (which is primarily a marker for macrophages but also found constitutively expressed on NK cells) and members of the B7 ligands family (CD86 and CD276/B7-H3). By contrast, only 7 genes were differentially expressed between mock and PD-1 KO in the HA HER2-28Z CAR-T cells, with the memory marker IL7R being the most relevant and upregulated in the PD-1 KO group (Fig. 4a, b, right panels). To provide deeper insight into the biological functions underlying the entire dysregulated gene expression signature pertaining to the comparison of mock and PD-1 KO LA CAR-T cells, we performed gene enrichment analysis[30,31]. As anticipated, we observed a significant enrichment in categories related to cytokine signaling, two of which were related to type 1 IFN signaling pathways (Fig. 4c and Supplementary Fig. 6a).

To better understand the intrinsic differences between LA and HA CAR-T cells that could explain the differential sensitivity towards the PD-1/PD-L1 axis, we directly compared the transcriptomic profile of both groups. We found that LA HER2-28Z CAR-T cells expressed preferentially genes associated with a more naïve phenotype such as TCF7, LEF1 and CD45RA[28,32,33], while HA CAR T-cells showed higher expression of the exhaustion-related transcription factor MAF (Supplementary Fig. 6b). We also compared the transcriptome of PD-1 KO LA HER2-28Z CAR-T cells with that of HA CAR-T cells, as both presented with similar anti-tumor activity. Regardless, PD-1 KO LA HER2-28Z CAR-T cells displayed a less exhausted phenotype with higher expression of BCL6, FOSB or TCF7 and lower expression of genes related to exhaustion such as IRF4, CTLA4, FAS or MAF (Supplementary Fig. 6c).

## PD-1 KO increases the polyfunctionality of LA but not HA HER2-28Z CAR-T cells

Since a number of transcriptional changes involved genes encoding cytokines and because the polyfunctionality of CAR-T cells had been previously correlated with improved clinical outcomes[34], we sought to assess whether genetic ablation of PD-1 increased the polyfunctionality of HER2-28Z CAR-T cells after antigen exposure. To this end, we performed a single-cell secretome analysis of low and high affinity HER2-28Z CAR-T cells after antigen exposure by using the Adaptive Immune cytokine panel (Isoplexis). We first observed in 3D t-SNE analysis that mock and PD-1 KO LA CAR-T cells segregated in separated clusters, revealing key functional differences between the groups (Fig. 5a). By contrast, in HA CAR-T cells, mock and PD-1 KO groups did not cluster separately, suggesting a more homogenous functional profile (Fig. 5b). These results are in agreement with the gene expression analysis (Fig. 4). In a deeper analysis, PD-1 KO in the LA HER2-28Z CAR-T cells showed trends towards increased percentages of cells producing two or more cytokines both in CD4$^+$ and CD8$^+$ T cell populations (Fig. 5c). We also measured the polyfunctional strength index (PSI), which accounts for levels and functional classification of the secreted cytokines. This analysis revealed an overall PSI increase by PD-1 disruption in LA HER2 CAR-T cells and a predominance of effector cytokines in all groups (Fig. 5e). In contrast, knocking out PD-1 in HA HER2-28Z CAR-T cells did not significantly increase T-cell polyfunctionality or PSI,

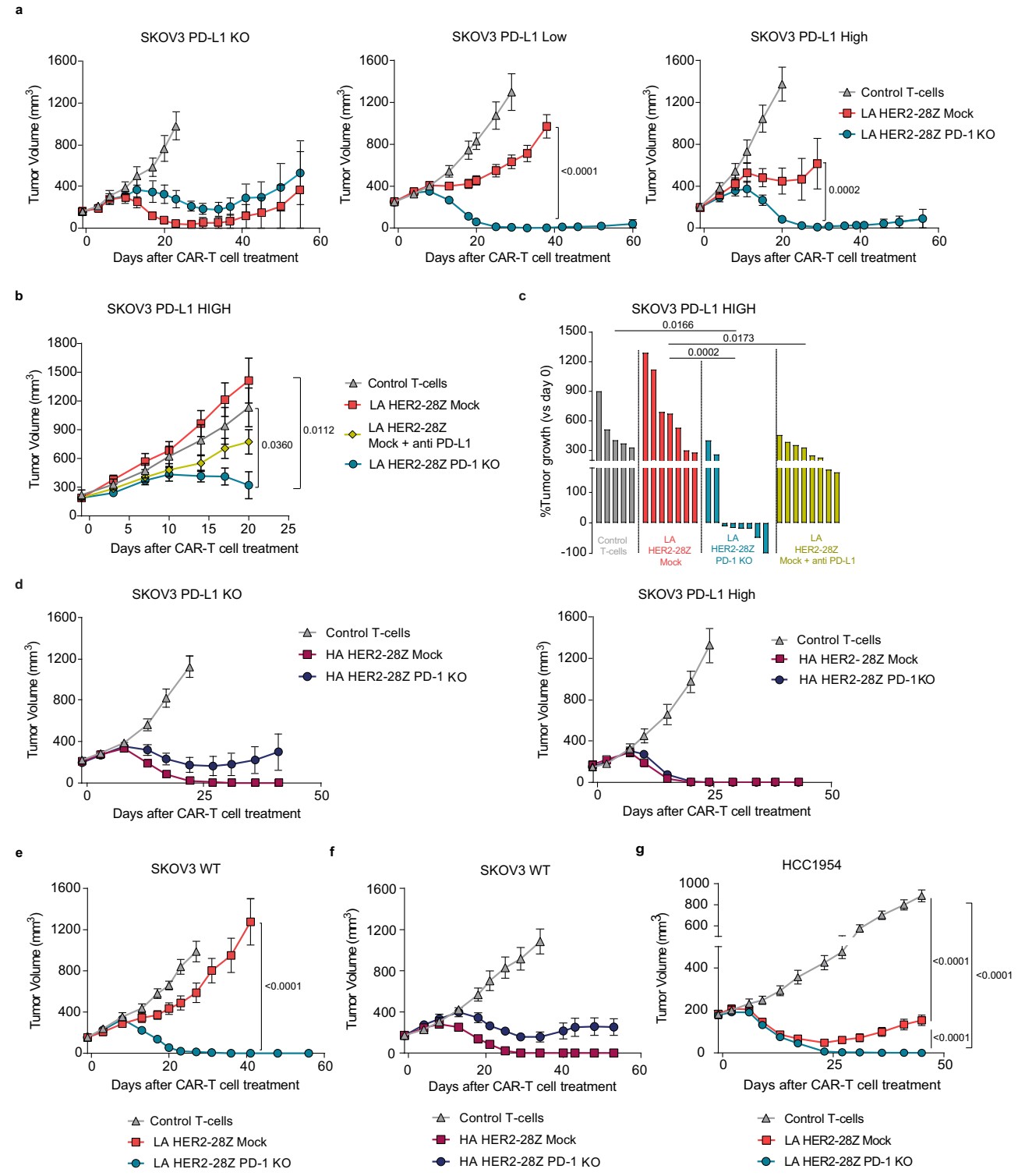

implying a less profound impact of PD-1 disruption (Fig. 5d, f). In addition, single-cell secretome analysis suggested a reduced expression of regulatory cytokines such as IL-13, IL-22 or IL-4 by LA PD-1 KO CAR-T cells, while PD-1 ablation did not alter the expression of these cytokines in HA CAR-T cells (Supplementary Fig. 7a).

To confirm the results obtained in the polyfunctionality study, we performed an intracellular cytokine staining (ICS) assay. While the PMA-Ionomycin-treated groups exhibited consistent outcomes across all experimental conditions (Supplementary Fig. 7b, c), PD-1 KO

enhanced the frequency of CAR-T cells concurrently releasing IFN-γ and TNF-α during co-culture with HER2+ tumor cells in the context of LA CAR-T cells. Conversely, no discernible effect between mock and PD-1 KO groups on HA CAR-T cells was observed (Fig. 5g, h). Of note, frequencies of IFN-γ⁺TNF-α⁺ T cells in mock groups from LA and HA CAR-T cells were comparable.

Altogether, the upregulation of genes associated with T-cell activation alongside the augmented polyfunctionality could contribute to the heightened anti-tumor efficacy observed in LA HER2-28Z PD-1 KO.

**Fig. 2 | PD-1 KO restores LA HER2-28Z CAR-T cell function in vivo but does not affect HA HER2-28Z CAR T-cells. a** Tumor measurements of NSG mice bearing SKOV3 tumors expressing indicated PD-L1 densities and treated with 3–4 × 10⁶ control T-cells, mock or PD-1 KO LA HER2-28Z CAR⁺-T cells (n = 8 for SKOV3 PD-L1 KO and PD-L1 High; n = 8, 13 or 12 for control, mock and PD-1 KO groups, respectively, for SKOV3 PD-L1 Low). **b, c** NSG mice bearing SKOV3 PD-L1 High tumors were treated with 3 × 10⁶ control T-cells (n = 5), mock (n = 7), mock + anti PD-L1 antibody (n = 7) or PD-1 KO (n = 8) LA HER2-28Z CAR⁺-T cells. **b** Tumor measurements and (**c**) percentage of tumor growth indicated as the change in tumor volume on day 20 versus baseline is shown. **d** Tumor measurements of NSG mice bearing SKOV3 tumors expressing indicated PD-L1 densities and treated with 3–4 × 10⁶ control T-

cells, mock or PD-1 KO HA HER2-28Z CAR⁺-T cells (n = 8, 13 and 12 for control, mock and PD-1 KO groups, respectively, for SKOV3 PD-L1 KO; n = 8 for SKOV3 PD-L1 High). Tumor measurements of NSG mice bearing SKOV3 wild type tumors treated with 3 × 10⁶ control T-cells, mock or PD-1 KO HER2–28Z CAR⁺-T cells of (**e**) LA (n = 8 for control and n = 10 for mock and PD-1 KO groups) or (**f**) HA (n = 8 for all groups). **g** Tumor measurements of NSG mice bearing HCC1954 tumors treated with 3 × 10⁶ control T-cells (n = 9), mock (n = 10) or PD-1 KO (n = 10) LA HER2-28Z CAR⁺-T cells. Data in (**a**, **b**) and (**d**–**g**) are represented as mean tumor volume ±SEM and n indicates tumors per group. p values by (**c**) one-way ANOVA with Tukey post-hoc test or (**a**, **b**, **d**–**g**) two-way ANOVA with Tukey's multiple testing correction are indicated. Source data are provided as a Source Data file.

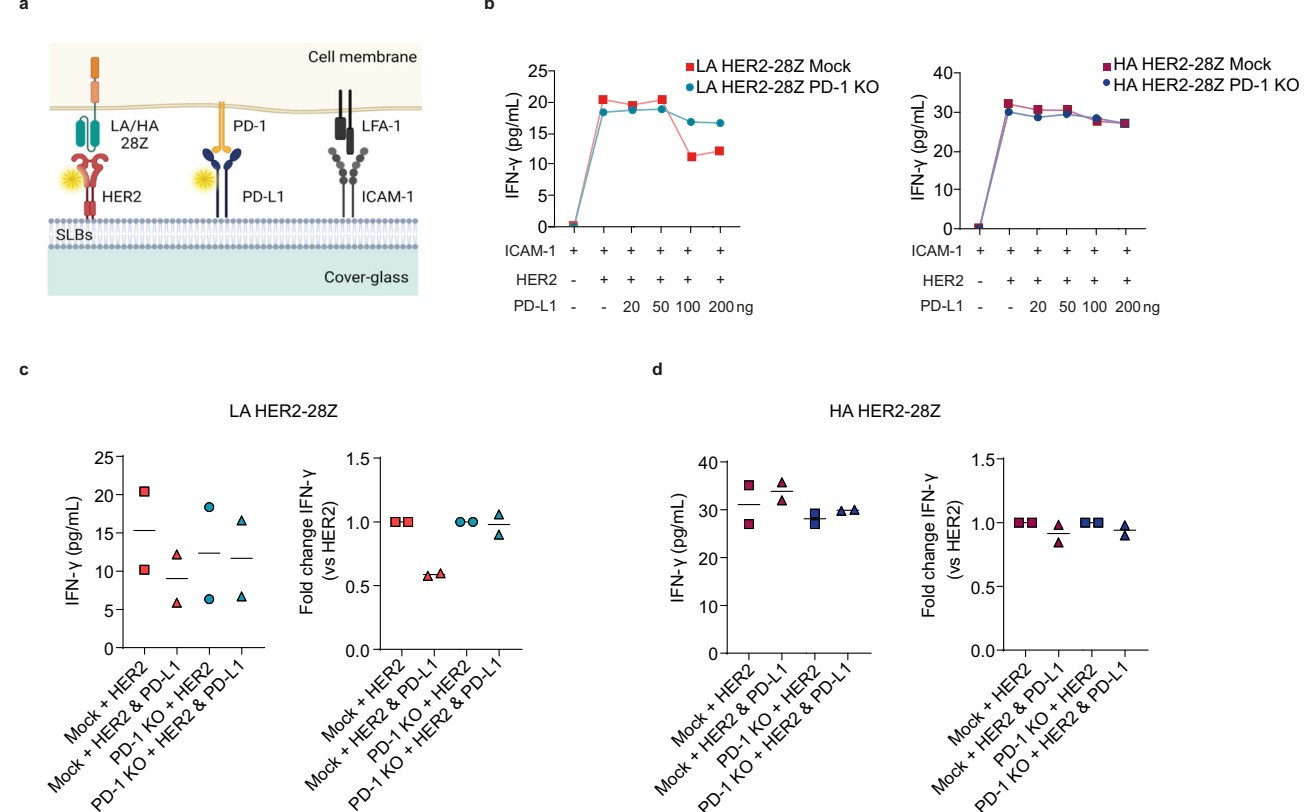

**Fig. 3 | HA HER2-28Z CAR-T cells are more resistant to inhibition by increasing amounts of PD-L1 in a protein-functionalized planar glass SLB system.**
**a** Schematic representation of an SLB featuring fluorescently labeled proteins (HER2 and PD-L1) and ICAM-1. Created with Biorender.com. **b** IFN-γ production by HER2-28Z mock and PD-1-KO LA (left panel) or HA (right panel) CAR-T cells after 24 h of co-culture with SLBs containing increasing concentrations of PD-L1. HER2-

28Z mock and PD-1-KO (**c**) LA or (**d**) HA CAR-T cells were co-cultured for 24 h with SLBs containing either HER2 alone (2 ng) or HER2 along with PD-L1 (200 ng). IFN-γ secretion as measured by ELISA is represented as absolute levels (left panel) or fold change of the HER2 + PD-L1 condition compared to HER2 alone (right panel). Data in (**b**) is representative of two different donors and in (**c**, **d**) is shown as mean of two different donors (n = 2). Source data are provided as a Source Data file.

Remarkably, the witnessed lack of distinguishable transcriptional changes at the transcriptomic level and in polyfunctionality supports the notion that PD-1 KO does not significantly impact the functional properties of HA HER2-28Z CAR-T cells.

## LA PD-1 KO display a safer toxicity profile as compared to HA HER2-28Z CAR-T cells
Using CAR-T cells resistant to the inhibition by the PD-1/PD-L1 axis may be an attractive strategy for the treatment of solid tumors. However, safety concerns arise when targeting tumor associated antigens using a high affinity CAR, as it may exhibit poor discrimination between tumor and healthy tissues expressing lower levels of the target antigen. To address this concern, we established co-cultures of LA or HA HER2-28Z CAR-T cells with or without PD-1 KO and a panel of human primary healthy cells including Epidermal Keratinocytes (NHEK), Renal Epithelial Cells (HREpC), Pulmonary Artery Endothelial Cells (HPAEC) and

Pulmonary Artery Smooth Muscle Cells (HPASMC), all of which have been reported to express low but detectable HER2 densities[20]. Both LA and HA HER2-28Z CAR-T cells demonstrated comparable reactivity against a control cancer cell line expressing high HER2 levels (Supplementary Fig. 8a–e). However, only HA CAR-T cells were activated in response to co-culture with healthy cells as evidenced by increased production of CD107-α, IFN-γ and IL-2 (Fig. 6a, b, c, and Supplementary Fig. 8f, respectively), raising safety concerns. Of note, PD-1 KO did not exacerbate the reactivity of LA HER2-28Z CAR-T cells against primary cells from healthy tissues, which showed a toxicity profile similar to non-tumor specific control T cells.

## Target antigen densities and CAR expression play a role in determining sensitivity to PD-L1
We then investigated how target antigen densities influence the heightened resistance of HA CAR-T cells to the PD-1/PD-L1 axis. We

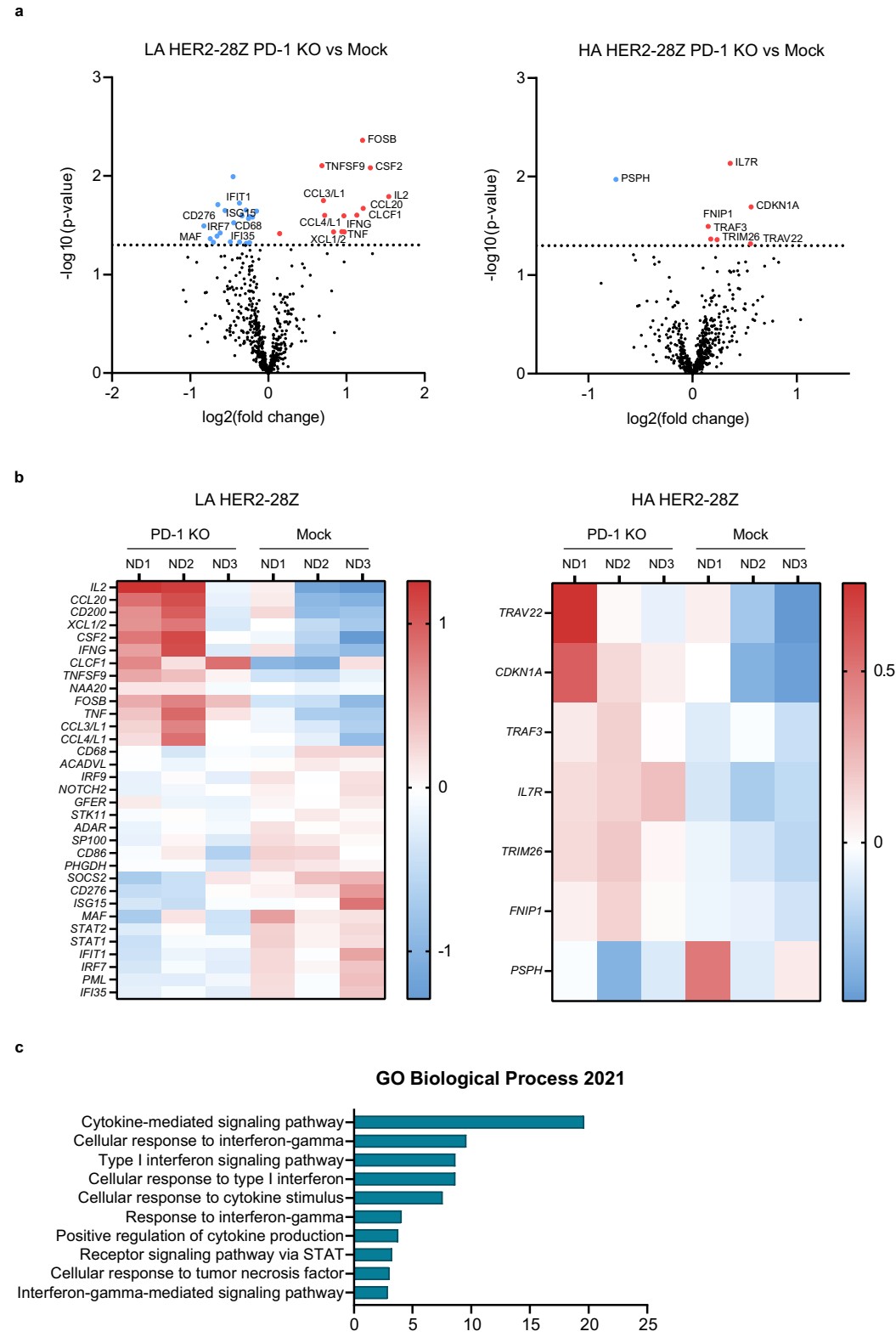

**a**

**b**

**c** GO Biological Process 2021

hypothesized that HA CAR-T cells might become susceptible to this inhibitory pathway under conditions of low antigen densities.

To explore this, we took advantage of the lipid bilayer model outlined in Fig. 3a to titrate down HER2 densities while maintaining constant high levels of PD-L1. In this controlled environment, HA CAR-T cells remained unaffected by PD-L1, as indicated by comparable levels of IFN-γ released by mock CAR-T cells across all HER2 conditions.

Of note, at the lowest antigen levels, mock and PD-1 KO exhibited similar behavior, while as HER2 levels increased, PD-1 KO appeared to have a detrimental effect (Fig. 7b). In the LA setting, PD-1 KO conferred an advantage to CAR-T cells at all antigen density conditions tested (Fig. 7a).

Next, we employed a cellular model based on a triple-negative breast cancer cell line, MDA-MB-468, engineered to express either low

**Fig. 4 | Differential transcriptomic response of HA and LA HER2-28Z CAR-T cells to PD-1 KO.** Transcriptomic analysis of mock and PD-1 KO LA or HA HER2-28Z CAR-T cells was performed after stimulation with SKOV3 WT tumor cells for 48 h. **a** Volcano plots of differential expression between mock and PD-1 KO in LA (left panel) or HA (right panel) HER2-28Z CAR-T cells. Red dots represent genes upregulated in PD-1 KO vs mock, blue dots represent genes downregulated genes in PD-1 KO vs mock and black dots represent genes not differentially expressed. Differentially expressed genes are annotated. The horizontal line is at an adjusted $p$ value of 0.05. **b** Heat map of differential expression between mock and PD-1 KO in LA (left panel) or HA (right panel) HER2−28Z CAR-T cells. **c** Gene Ontology (GO) Biological process of differentially expressed genes between PD-1 KO and mock LA HER2-28Z CAR-T cells. Data in (**a**–**c**) is represented as mean of $n = 3$ donors. In (**a**, **b**) $p$ value thresholds ($p < 0.05$) were derived from Rosalind and adjusted using the Benjamini−Hochberg method. In (**c**) $p$ value thresholds ($p < 0.05$) were derived from Enrichr by a Fisher exact test and adjusted using the Benjamini−Hochberg method. All tests were two-sided. Source data are provided as a Source Data file.

or high levels of HER2 along with constitutive high levels of PD-L1 (Supplementary Fig. 9a). Consistent with our observations in the SKOV3 model, PD-1 KO provided an advantage to LA CAR-T cells when HER2 levels were high. However, under conditions of high antigen densities, HA HER2-28Z CAR-T cells did not benefit from PD-1 KO (Fig. 7c, e–f). Conversely, in co-culture with HER2-low cells, PD-1 KO conferred an advantage to HA CAR-T cells under certain settings, reaching statistical significance in terms of increased percentage of polyfunctional T cells producing both IFN-γ and TNF-α (Fig. 7e, g and Supplementary Fig. 9c, d) and IL-2 secretion (Supplementary Fig. 9b), but not in IFN- γ as measured by ELISA (Fig. 7d). In line with toxicity results in Fig. 6, LA HER2-28Z CAR-T cells did not exhibit reactivity in low antigen conditions.

Based on these results, we hypothesized that PD-L1-mediated inhibition could potentially be overcome at a certain threshold of T-cell activation, and that this could also be achieved by utilizing T-cell products with high percentage of CAR transduction. To validate this hypothesis, we conducted studies with T-cell products containing more than 75% CAR⁺ T cells (High CAR) as compared to products containing 50-65% CAR⁺ T cells (Low CAR) (Fig. 7h). In this scenario, the advantage provided by PD-1 KO in the LA CAR-T cells was lessened (Fig. 7i), similar to our observations in the HA setting (Fig. 7j).

Overall, our results demonstrate that although CAR affinity is pivotal in determining sensitivity of CAR-T cells to PD-1/PD-L1 axis, other factors such as antigen density and CAR expression levels may also play a role.

### Advantages of PD-1 KO do not apply uniformly across different CAR constructs

To determine whether our observations with CD28-based HER2 CAR-T cells could be applied to CARs containing different co-stimulation domains we first interrogated CARs targeting HER2 with either LA or HA and containing ICOS as a co-stimulatory domain in vivo (Fig. 8a)[35]. As shown in Fig. 8b, we found that PD-1 ablation enhanced the anti-tumor efficacy of LA but not HA HER2 CAR-T cells in mice containing SKOV3 wild-type tumors, consistent with our earlier findings in CD28-based CAR-T cells. Since 4-1BB is a clinically relevant co-stimulatory domain, we also explored how the PD-1/PD-L1 axis impacted 4-1BB co-stimulated LA CAR-T cells (HER2-BBZ, Fig. 8a). We observed that neither PD-1 KO nor PD-1/PD-L1 blockade by using antibodies increased cytokine secretion in vitro (Fig. 8c) or anti-tumor effect in vivo (Fig. 8d). Higher resistance of 4-1BB-based CARs to PD-L1 was also confirmed in vitro by using CAR-T cells targeting mesothelin also with LA (Mesothelin-BBZ, Fig. 8a and Supplementary Fig. 10c). Differential CAR expression was ruled out as a potential reason for the differing sensitivity to PD-L1-mediated inhibition among constructs with distinct co-stimulatory domains, as ICOS-based CARs, despite being expressed at lower levels as compared to CD28, were still sensitive to PD-1/PD-L1 axis. In contrast, 4-1BB-based CARs exhibited comparable expression levels to CD28 but demonstrated greater resistance to inhibition by PD-L1 (Supplementary Fig. 10a, b). These results can be explained, at least in part, by the lower expression levels of PD-1 when compared to that of CD28- or ICOS co-stimulated CARs (Supplementary Fig. 10d) and are in line with previous studies showing that 4-1BB-based CARs are less sensitive to PD-1 mediated inhibition than CD28-based CARs[8,36].

Finally, we explored if the higher sensitivity to PD-1/PD-L1 inhibitory pathway observed in LA HER2 CAR-T cells was also maintained in CAR-T cells targeting other antigens with LA. To this end we explored the effects of PD-1 KO in CAR-T cells targeting mesothelin with LA and containing the CD28 intracellular domain (Mesothelin-28Z, Fig. 8a)[37]. Interestingly, we observed that in T-cells expressing an anti-mesothelin LA CAR, blocking the PD-1/PD-L1 axis through genetic disruption or with the use of PD-1 or PD-L1 blocking antibodies resulted both in elevated in vitro cytokine secretion (Fig. 8e) and enhanced in vivo anti-tumor effect (Fig. 8f), confirming results obtained with LA HER2-28Z CAR-T cells.

## Discussion

The findings reported herein unveil CAR affinity as a factor modulating the sensitivity of CAR-T cells to PD-1/PD-L1 axis. By using a preclinical model of tumor cells expressing varying PD-L1 densities, we found that low affinity CAR-T cells are more sensitive to PD-L1-mediated inhibition as compared to high affinity CARs. Accordingly, PD-1 disruption only impacted positively on the functionality of low affinity CAR-T cells, while high affinity CAR-T cells remained unaltered.

The potential of disrupting PD-1/PD-L1 signaling as a strategy to overcome PD-L1-mediated T cell suppression and to boost the therapeutic index of CAR-T cells has been widely discussed. While most works report increased functionality of PD-1-ablated CAR-T cells[8–17], some others suggest that PD-1 disruption accelerates T cell exhaustion and impairs long-term T cell persistence[18,19]. We identified a lack of consistency within the different articles in terms of the tumor models used, ranging from cell lines engineered to constitutively express high levels of PD-L1 to cell lines expressing physiological levels of PD-L1 in response to CAR-T cell activation. This might in part explain discrepancies in the reported results. Arising from this observation, and by the fact that currently available preclinical models often fail to predict clinical outcomes[38], the first goal of our project was the generation of robust preclinical models for systematic interrogation of different CARs. Our cellular-based model demonstrated to be representative of the range of physiological PD-L1 expression levels observed in different tumor cell lines across cancer types. As an additional preclinical model to validate our findings, we developed glass-supported lipid bilayers (SLBs) mimicking the target cell membrane but containing precisely defined amounts of surface proteins, which supposed a valuable tool that allowed the determination of exact PD-L1 amounts required for T cell inhibition[22].

The most intriguing observation of our work, as observed in both preclinical models, was that HA CAR-T cells were intrinsically more resistant to PD-L1-mediated inhibition as compared to their LA counterparts. Although we primarily used affinity tuned HER2-specific CARs of low and high affinity[20] based on CD28 as co-stimulation domain, similar observations were made for CARs targeting FRβ or mesothelin and ICOS-co-stimulated CARs, suggesting that this effect is not unique for a specific targeted antigen or co-stimulatory domain. More in-deep gene expression and single-cell polyfunctionality analysis after exposure to tumor cells expressing HER2 and physiological PD-L1 levels revealed that whether PD-1 KO induced a significant change in the LA CAR-T cell population at the transcriptomic and functional level, HA CAR-T cells after PD-1 KO remained similar to the mock-electroporated CAR-T cells. In this work, we emphasize CAR affinity as a central factor

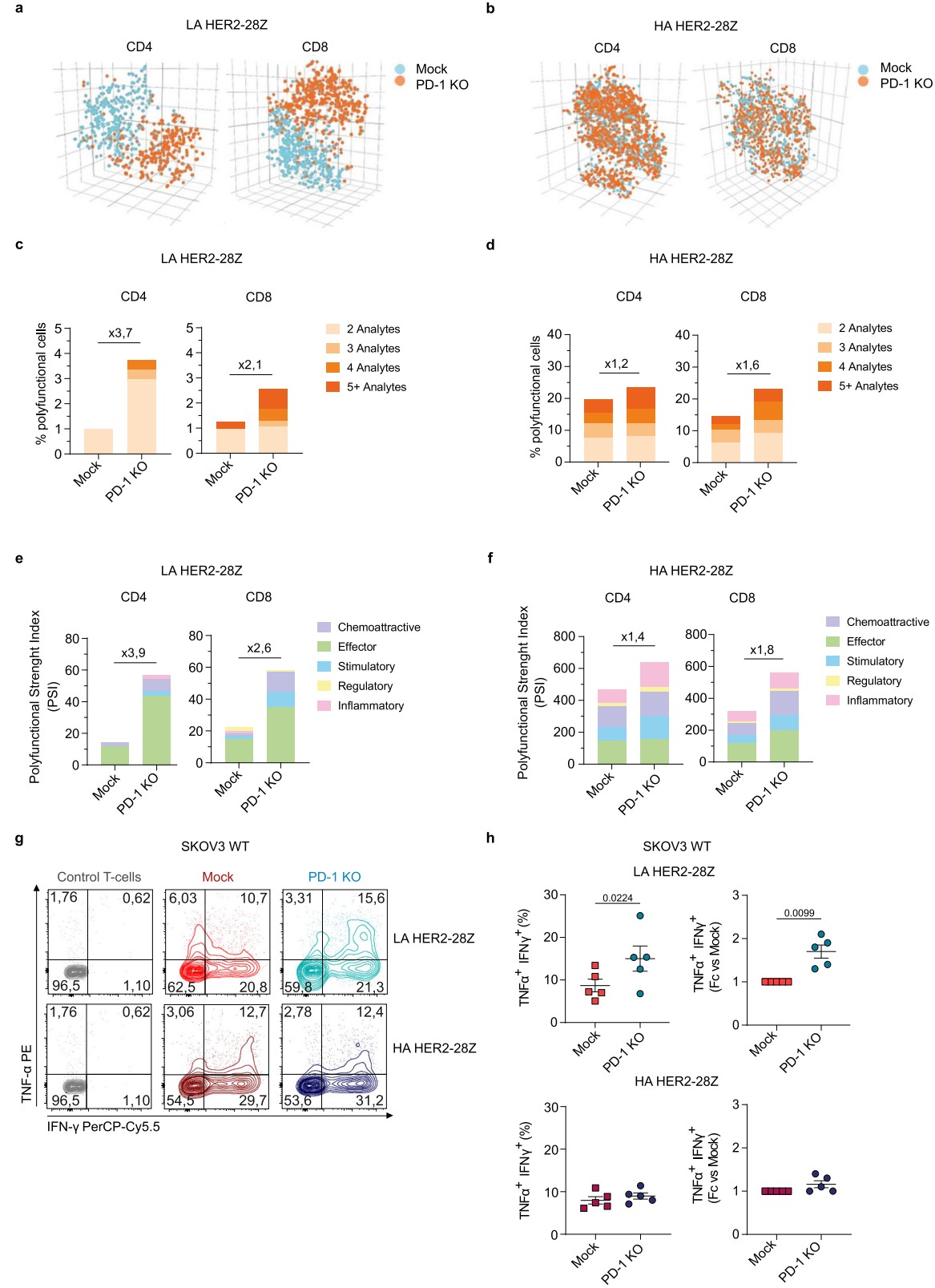

influencing sensitivity to PD-1/PD-L1 axis. Beyond this, we also explored the implications of both target antigen densities and CAR expression frequencies. Intriguingly, we observed that the enhanced resistance of HA CAR-T cells to PD-L1 was attenuated in the presence of low levels of CAR antigen. Conversely, the use of products with higher CAR frequencies appeared to mitigate the sensitivity of LA CAR-T cells to PD-L1-mediated suppression. These findings lead us to postulate that

inhibitory effects of PD-L1 on T cells can be overcome when T cell activation reaches a certain threshold. According to our results, this threshold of activation can be attainable not only through the utilization of a high affinity CAR but also by the presence of high CAR frequencies. However, in principle this latter strategy lacks the potential of being translated into the clinical setting due to risk of genotoxicity, given the higher amounts of vector required. Moreover, the use of

**Fig. 5 | Polyfunctional profiling of PD-1 KO LA and HA HER2-28Z CAR-T cells.** Single-Cell Adaptive Immune panel (Isoplexis) of mock and PD-1 KO LA and HA HER2-28Z CAR-T cells after co-culture with SKOV3 tumor cells for 24 h (E:T = 1:3). Three-dimensional t-SNE plots of (**a**) LA and (**b**) HA mock (blue) and PD-1 KO (orange) HER2-28Z CAR-T cells by differentiating them based on their cytokine functional differences. Frequencies of polyfunctional cells of mock and PD-1 KO (**c**) LA and (**d**) HA HER2-28Z CAR-T cells. Polyfunctionality Strength Index (PSI) of mock and PD-1 KO (**e**) LA and (**f**) HA HER2-28Z CAR-T cells. Fold-change values for PD-1 KO versus mock are indicated. **g** Representative flow cytometry plots of intracellular cytokine staining for TNF-α and IFN-γ in mock and PD-1 KO LA and HA HER2−28Z CAR-T cells after co-culture with SKOV3 tumor cells for 24 hours (E:T = 1:3) (gated on live/CD45⁺). **h** Frequencies of IFN-γ⁺TNF-α⁺ T-cells represented as absolute numbers and fold change of PD-1 KO versus mock are shown. In (**a**–**f**), data is shown as mean from $n = 2$ donors. In (**h**), data is pooled from four independent experiments where each dot represents CAR-T cells generated from a different donor ($n = 5$), and is represented as mean ± SEM. $p$ values by a two-tailed paired $T$ test for absolute numbers or by two-tailed one-sample $T$ test for fold-change graphs are indicated. Source data are provided as a Source Data file.

CAR-T cell products with high CAR expression has been correlated to worse clinical responses due to accelerated T cell exhaustion[39]. Interestingly, in apparent contradiction to this, one of the most successful CAR-T trials in solid tumors to date in term of response rates, utilized products with >70% CAR transduction, potentially contributing to the success[2,40]. Overall, our results highlight the complexity of CAR-T cell activity regulation, involving numerous interplaying factors.

The formation of a productive immune synapse (IS) is crucial to achieve optimal T-cell activation, and adhesion molecules are crucial players in IS formation[41]. One of those is ICAM-1, which has been shown to be instrumental for CAR-T cell effector function[42,43], and its upregulation has also been implicated in resistance to PD-1/PD-L1 pathway[44], all in an IFN-γ-dependent manner. One can hypothesize that as HA CAR-T cells release higher levels of IFN-γ upon co-culture with tumor cells as compared to LA CAR-T cells, they can induce increased upregulation of ICAM-1 and therefore, increased resistance to PD-L1. Recently, another adhesion molecule, CD56, has been reported to play a role in CAR-T cell effectiveness in triple inhibitory receptor-resistant CAR-T cells (including PD-1 knockdown)[45]. Considering these works, we cannot rule out the potential implication of alternative adhesion molecules in the resistance to PD-L1-mediated inhibition of HA CAR-T cells.

Concerns regarding accelerated T-cell exhaustion following PD-1 ablation have been raised in previous studies, both in the context of CAR-T cells for cancer treatment[18,19] and in virus-specific T cells in chronic infections[46]. Our transcriptomic analysis did not reveal a more exhausted phenotype of LA PD-1 KO CAR-T cells but rather the opposite. In fact, PD-1 KO CAR-T cells presented hallmarks of less exhausted T cells as compared to mock CAR-T cells. In this regard, we found that 11 out of 20 genes that were downregulated in PD-1 KO LA CAR-T cells were genes involved in the type I and II IFN signaling pathways, including IRF7 which is the main transcription factor regulating type I IFN pathway. Although IRF7 induction can potentiate CAR-T cell activation and induce antitumor activity[47], in certain contexts, type I IFN signaling can also orchestrate T cell immunosuppression[48,49] and induce apoptosis on CAR-T cells[50]. Recent papers identified chronic type I IFN signaling regulated by IRF7 to potentiate CAR-T cell dysfunction[27] and to be predictive of poor CAR-T cell persistence in pediatric acute lymphoblastic leukemia (ALL) patients[28]. By contrast, PD-1 KO CAR-T cells in our experiments expressed higher levels of FosB, a transcription factor that is decreased in exhausted T cells in chronic viral infection[24] while increased in CAR-T cell products from responding patients as compared to non-responders[25]. However, it is important to note that our transcriptomic data was obtained after a single antigen stimulation in vitro, and further exploration of what would happen in the context of repeated stimulations might be required.

In broader terms, by using our preclinical model, we also observed that PD-1 KO does not increase the antitumor efficacy of LA CARs co-stimulated with 4-1BB, contrary to CARs featuring CD28 or ICOS as co-stimulatory domains. Our results are in line with the previously described by others[8,36] and might be attributed in part to the PD-1 low phenotype but also to the distinct pathway that 4-1BB signal through as compared to CD28 and ICOS. It is well-established that PD-1 activation by PD-L1 primarily suppresses T-cell function through the deactivation of CD28 signaling, suggesting the central role played by co-stimulatory pathways within the context of PD-1 therapy[51,52].

Of note, it is relevant to highlight that even in the cases where PD-1 genetic deletion does not provide an advantage (i.e., HA and 4-1BB co-stimulated CAR-T cells) it never decreases CAR-T cell functionality in our hands. This observation offers the potential to repurpose PD-1 as a site for targeted integration of therapeutic transgenes, capitalizing the kinetics of PD-1 expression after antigen encounter to restrict transgene expression to the tumor microenvironment while simultaneously disrupting PD-1[53,54].

In terms of clinical translation of our findings, the use of a high affinity CAR might be preferable as it exhibits greater efficacy and resistance to PD-L1-mediated inhibition without additional modifications. However, increased resistance to PD-L1 might come at the price of increased T-cell exhaustion and diminished safety. Our transcriptomic data supports the notion of that HA CAR-T cells may be more prone to exhaustion. In the literature, a recent study demonstrated less exhausted and apoptotic phenotype and greater persistence of CAR-T cells targeting GPC3 with low affinity as compared to their high affinity counterparts[55]. In the same line, a CAR targeting CD19 with lower affinity than commercial products demonstrated greater persistence in preclinical mouse models and patients in a clinical study[56]. Regarding safety concerns, a serious event occurred in the context of HER2-targeting CAR therapy where the use of a HA CAR (based on the scFv 4D5, as employed in our study) led to a fatal outcome in a patient with colon cancer metastatic to the lungs and liver. This was attributed to the high doses of CAR-T cells administered and to the potential CAR-mediated recognition of low levels of HER2 on lung epithelial cells[57]. Our findings evidence a more favorable toxicity profile of LA PD-1 KO as compared to HA HER2-28Z CAR-T cells. This underscores the necessity for caution and thorough investigation when employing HA CARs, emphasizing their potential for unintended activation in the presence of healthy cells expressing lower levels of the target antigen. Interestingly, in terms of efficacy, an analysis of available data from solid tumor CAR-T trials correlating clinical responses to CAR affinity concluded that the use of CARs targeting their antigens with moderate affinity led to best clinical responses as compared to high affinity CARs[58].

Regarding the methodology employed for PD-1/PD-L1 disruption, our study utilized CRISPR/Cas9 to knockout *pdcd1* as a proof of concept. This method is highly efficient, ensures sustained PD-1 blockade, circumvents toxicities associated with systemic PD-1 blockade and has demonstrated safety in the clinics[59,60]. However, we also provide data showing the feasibility of CAR-T cell combination with PD-1/PD-L1 blockade antibodies. Combining CAR-T cells with immune checkpoint antibodies offers other advantages such as more precise and flexible dosing regimen, eliminates the need for further genetic modifications on T cells, and can impact both endogenous T cells and CAR-T cells. In fact, combination of CAR-T cells with PD-1 blocking antibodies has also been explored in clinical trials[61]. This broadens the scope of therapeutic possibilities, emphasizing the adaptability of our findings to diverse PD-1 disruption approaches in the pursuit of enhanced CAR-T cell therapy.

In conclusion, our study reveals that CAR affinity plays a role in determining the sensitivity of CAR-T cells to T-cell inhibition mediated by the PD-1/PD-L1 axis. We have demonstrated that HA CAR-T cells

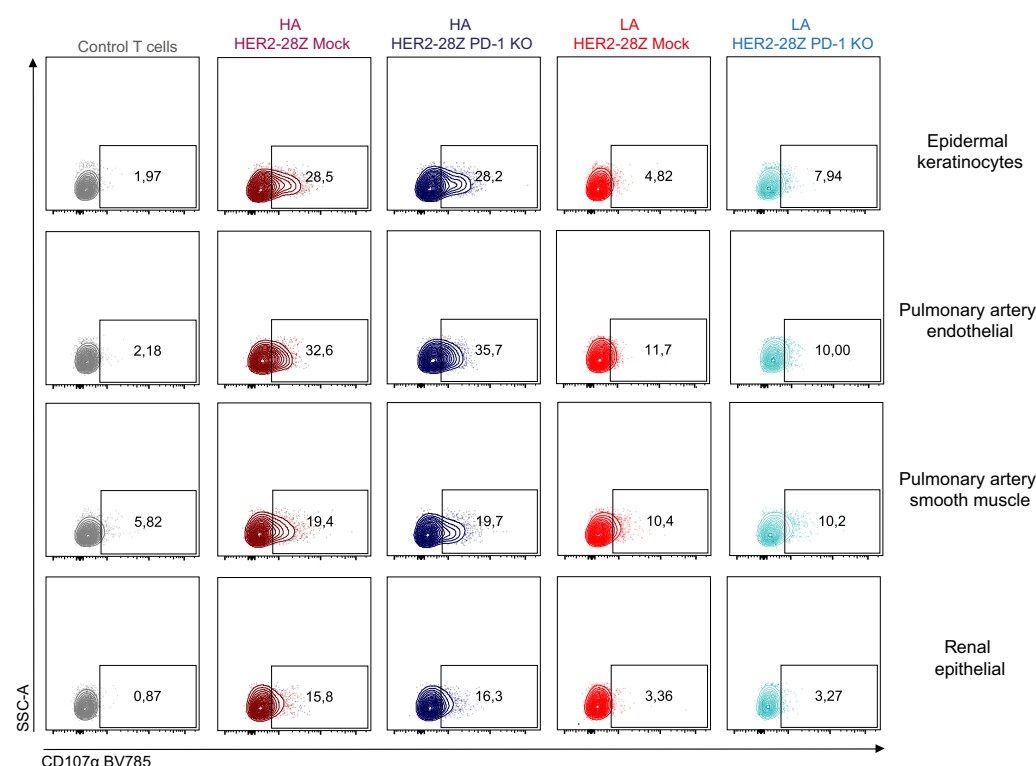

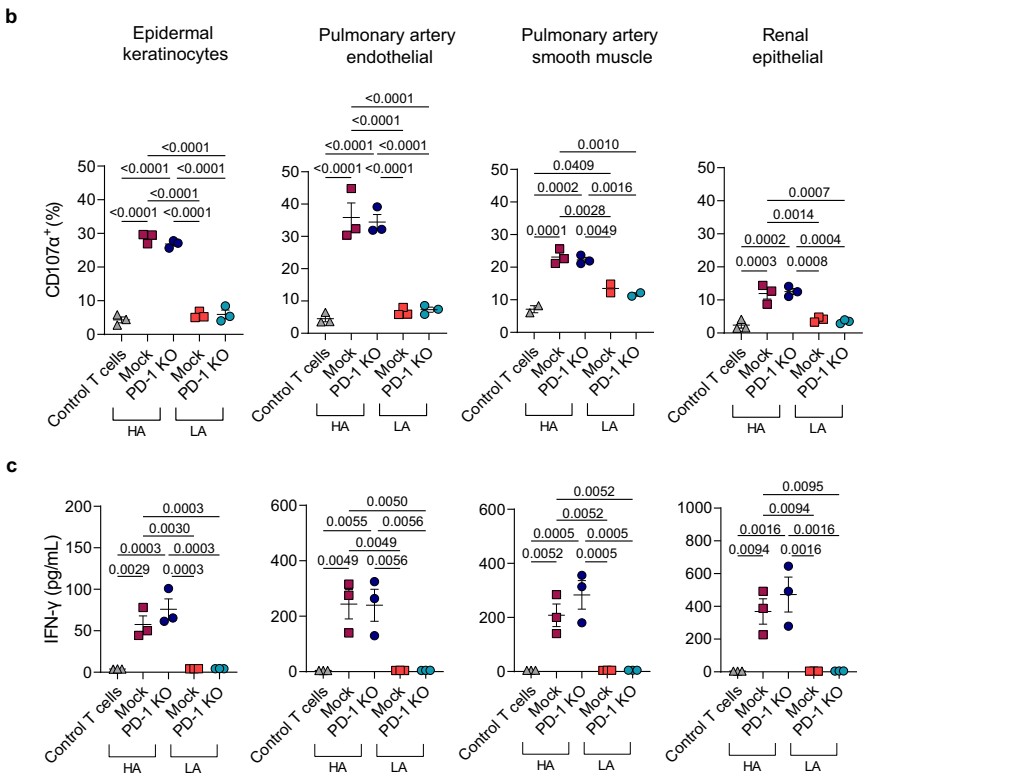

**Fig. 6 | HA HER2-28Z CAR-T cells demonstrate reactivity against a panel of primary healthy cells while LA CAR-T cells do not.** Mock or PD-1 KO HER2-28Z CAR-T cells of LA or HA were co-cultured with a panel of human primary cells. **a**, **b** CD107-α degranulation marker was measured after 6 h of co-culture (E:T = 1:1). **a** Representative flow cytometry plots and (**b**) percentage of cells producing CD107-α (gated on live/CD45⁺) are shown. **c** IFN-γ production by HER2-28Z CAR-T cells after 24 h of co-culture (E:T = 3:1) as quantified by ELISA. Data in (**b**, **c**) are plotted as mean ± SEM ($n$ = 3 donors). $p$ values by one-way ANOVA with Tukey's multiple testing correction are indicated. Source data are provided as a Source Data file.

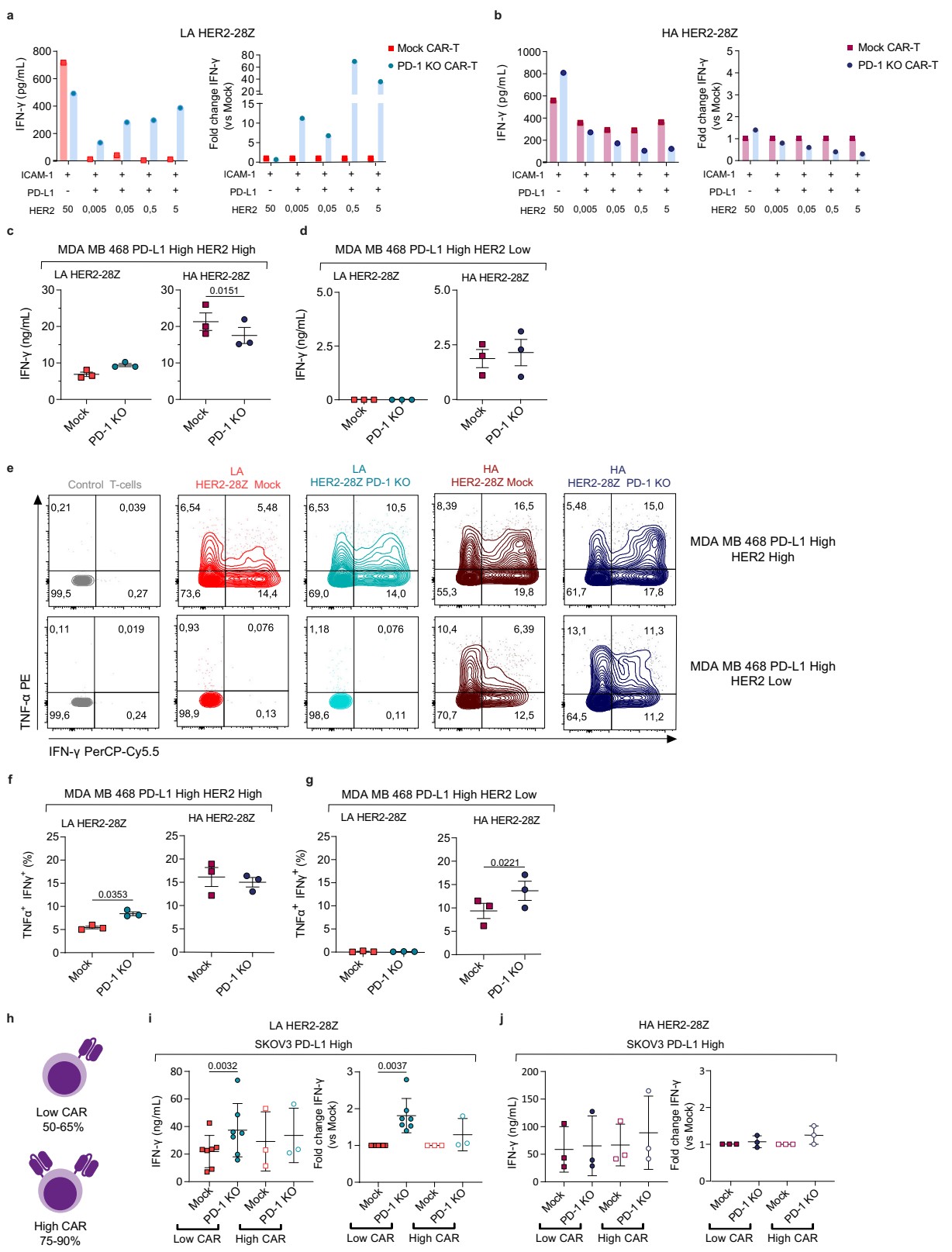

exhibit inherent resistance to PD-L1-mediated inhibition, whereas LA CARs are more susceptible to this suppression. In essence, these findings provide valuable insights into the design and optimization of CAR-T cells for enhanced effectiveness in the treatment of solid tumors, and particularly shed light on how to target the PD-1/PD-L1 axis more effectively in combination with the use of CAR-T cells as the field moves forward to clinical applications.

## Methods

### Study approval

Human T cells are isolated from buffy coats obtained from the Barcelona Public Blood and Tissue Bank. All samples are deidentified prior to receipt and no protected health information is transferred from the blood bank to our team or institution. Therefore, informed consent is not required from our side. Specific approval for this project was

**Fig. 7 | Role of target antigen and CAR expression in determining sensitivity to PD-L1.** IFN-γ production by mock or PD-1-KO (**a**) LA or (**b**) HA HER2-28Z CAR-T cells after 72 h of co-culture with SLBs containing either HER2 alone (50 ng) or increasing concentrations of HER2 along with PD-L1 (200 ng) as measured by ELISA. Data from one donor is represented as absolute levels (left panel) or fold change of IFN-γ by PD-1 KO versus mock HER2-28Z CAR-T cells (right panel). IFN-γ production by mock or PD-1-KO LA (left panel) or HA (right panel) HER2-28Z CAR-T cells after 24 h of co-culture with (**c**) MDA-MB-468 PD-L1 high HER2 high or (**d**) MDA-MB-468 PD-L1 high HER2 low (E:T = 3:1) as measured by ELISA. **e** Representative flow cytometry plots and frequencies of IFN-γ⁺TNF-α⁺ T-cells (gated on live/CD45⁺) of intracellular cytokine staining for TNF-α and IFN-γ in mock and PD-1 KO LA and HA HER2-28Z CAR-T cells after co-culture with MDA-MB-468 PD-L1 high HER2 high (**f**) or MDA-MB-468 PD-L1 high HER2 low (**g**) tumor cells for 24 h (E:T = 1:3). **h** Schematic

representation of criteria to discriminate between Low CAR and High CAR-T cell products. Created with Adobe Illustrator. IFN-γ production by mock or PD-1-KO (**i**) LA or (**j**) HA HER2-28Z CAR-T cells with either Low ($n = 7$ donors for LA and $n = 4$ for HA) or High ($n = 3$ donors) CAR frequencies after 24 h of co-culture with SKOV3 PD-L1 High tumor cells (E:T = 3:1). Data in (**c**, **d**, **f**, **g**) is represented as mean ± SEM ($n = 3$ donors) and $p$ values by a two-tailed paired $T$-test are indicated. Data in (**i**) and (**j**) is pooled from nine and five independent experiments, respectively, where each dot represents CAR-T cells generated from different donors (n) and represented as mean ± SEM for absolute levels (left panel) and fold change of PD-1 KO versus mock (right panel). $p$ values by a two-tailed paired $T$ test (for absolute levels) or a two-tailed one-sample $T$-test (for fold change) are indicated. Source data are provided as a Source Data file.

obtained from the local ethic committee (Comité de Ética de la investigación con medicamentos CEIm).

All mouse studies were performed under a protocol (184-20) approved by the Ethic Committee for Animal Experimentation (CEEA) of the University of Barcelona and Generalitat de Catalunya.

## Cell line culture
Details of all cell lines used in this study are listed in Supplementary Table 1. All cell lines were grown at 37 °C and 5% $CO_2$ and were regularly validated to be *Mycoplasma free* and authenticated in 2019 by IDEXX Bioanalytics using the Human 9-Marker STR Profile.

## Generation of cancer cell lines
SKOV3 cells were genome edited to delete CD274 (PD-L1) using the CRISPR-Cas9 system. Single guide targeting CD274 (A*U*U*UACUGU-CACGGUUCCCA) was synthetized by Synthego. Ribonucleoprotein (RNP) complexes were formed by mixing the sgRNA and the TrueCut™ Cas9 Protein v2 (ThermoFisher) at a ratio of 3:1 and incubated for 10–15 min at room temperature following the per manufacturer's protocol (ThermoFisher). RNP complexes where then added to $5 \times 10^6$ SKOV3 and the cells were electroporated with the following conditions: 1170 V, 30mseg and 2 pulses using the Neon transfection system (ThermoFisher). PD-L1 negative cell population was sorted by flow cytometry after treatment with IFN-γ to induce PD-L1 expression and to allow accurate selection of the PD-L1 negative population. The SKOV3 PD-L1 KO cell line was then transduced with lentiviral vectors expressing PD-L1 under different promoters: EF1α (high expression) and PGK100 (low). pCCL-EF1α-PD-L1 was synthetized by Genscript. To generate pCCL-PGK100-PD-L1, an already created plasmid in the lab pCCL-PGK100-HER2t and pCCL-EF1α-PD-L1 were digested with XbaI and SalI-HF (From NEB) to obtain pCCL-PGK100 and PD-L1 fragments. Then, the backbone with the promoters and the PD-L1 sequence were purified using QIAquick PCR Purification Kit (QIAGEN), ligated, and transformed in Stbl3 (ThermoFisher). Five days after transduction, tumor cells were stained with L/D aqua and PD-L1 APC antibodies, and PD-L1⁺ tumor cells were collected separately using FacsAriaII cell sorter (BD). Copy numbers of PD-L1 molecules on cell surface were estimated using the Quantibrite™ Beads PE Fluorescence Quantitation Kit (ref. 340495, BD) according to the instructions of the manufacturer. SKOV3 PD-L1 KO and SKOV3 PD-L1 high cell lines were further modified to express folate receptor beta (FRβ) by using a lentiviral vector expressing FRβ under EF1α promoter. pCCL-EF1α-FRβ was synthesized by Genscript. After transduction, tumor cells were stained with an anti-FRβ antibody and FRβ+ tumor cells were collected using a FacsAriaII cell sorter (BD). Triple negative breast cancer cells (MDA-MB-468) were transduced with lentiviral vectors expressing a truncated version of HER2 lacking the intracellular domain under different promoters: EF1α (high expression) and PGK100 (low). After transduction, tumor cells were stained with an anti-HER2 antibody, and HER2+ tumor cells were collected using a FacsAriaII cell sorter (BD). MDA-MB-468 HER2

low and MDA-MB-468 HER2 high were further modified to express PD-L1 under the control of EF1α, as detailed for SKOV3 cells.

## Preparation of SLBs
Supported lipid bilayers (SLBs) were prepared as previously described[23]. First, 1,2-dioleoyl-sn-glycero-3-[N(5-amino-1-carboxypentyl) iminodiacetic acid succinyl] (nickel salt) (DGS-NTA(Ni)) and 1-palmitoyl-2-oleoyl-sn-glycero-3-phosphocholine were dissolved in chloroform and mixed in a 1:50 molar ratio. The mixture was then dried under vacuum overnight and resuspended in degassed PBS. Sonication was performed under nitrogen until the suspension became clear. Nonunilamellar vesicles were pelleted through ultracentrifugation, and the clear supernatant was subjected to further centrifugation. The second supernatant was filtered and stored under nitrogen. Glass slides were cleaned for 15 min using plasma (Zepto, Diener Electronic). Cleaned slides were attached to the bottom of an 12-well Nunc Lab-Tek chamber (Thermo Fisher Scientific) with Picodent twinsil extrahart (Picodent) until the glue had solidified. The lipid vesicle suspension was diluted 1:20 with PBS and filtered, and 100 µL of the diluted suspension were added to each well to form a continuous SLB. Excess vesicles were removed by washing the chambers with PBS. H12-tagged proteins were added to the SLBs and incubated for 60 min in the dark at room temperature. Finally, the chambers were rinsed with PBS to remove unbound protein.

## CAR construction and lentiviral production
Single-chain variable fragments (scFv) used for targeting HER2 or FRβ with low or high affinity were previously described[20,21,62]. Similarly, scFv sequence of M11 (targeting mesothelin) was extracted from patent WO2015090230A1 (Human mesothelin chimeric antigen receptors and uses thereof)[37]. All CAR sequences (including the mentioned scFvs, signal peptide, CD8 hinge, CD28 or CD8 transmembrane regions and intracellular domains from CD28, 41BB or ICOS and CD3Z), were synthesized by BaseClear B.V. or Genscript and cloned into the third-generation lentiviral vector pCCL under the control of EF1α promoter[63,64]. Lentiviral vectors were produced after transfection of 293FT and tittered in Jurkat cells as previously described[35]. Briefly, 293FT cells were seeded at $10 \times 10^6$ in a total volume of 18 mL of medium in a p150 culture plate. Eighteen hours later, 293FT cells were transfected with 18 µg of pCCL transfer plasmid (containing CAR) and a pre-mixed packaging mix containing 15 µg of pREV, 15 µg of pRRE and 7 µg of pVSV using PEI® (Polysciences). The viral supernatant was harvested at 48 and 72 h post-transfection, 0.45 µm filtered, concentrated by LentiX as per manufacturer's protocol (Clontech) and frozen at −80 °C until use.

## Isolation, transduction, electroporation and expansion of primary human T lymphocytes
Human T cells were isolated from healthy donor buffy coats obtained from the Barcelona Public Blood and Tissue Bank and expanded as

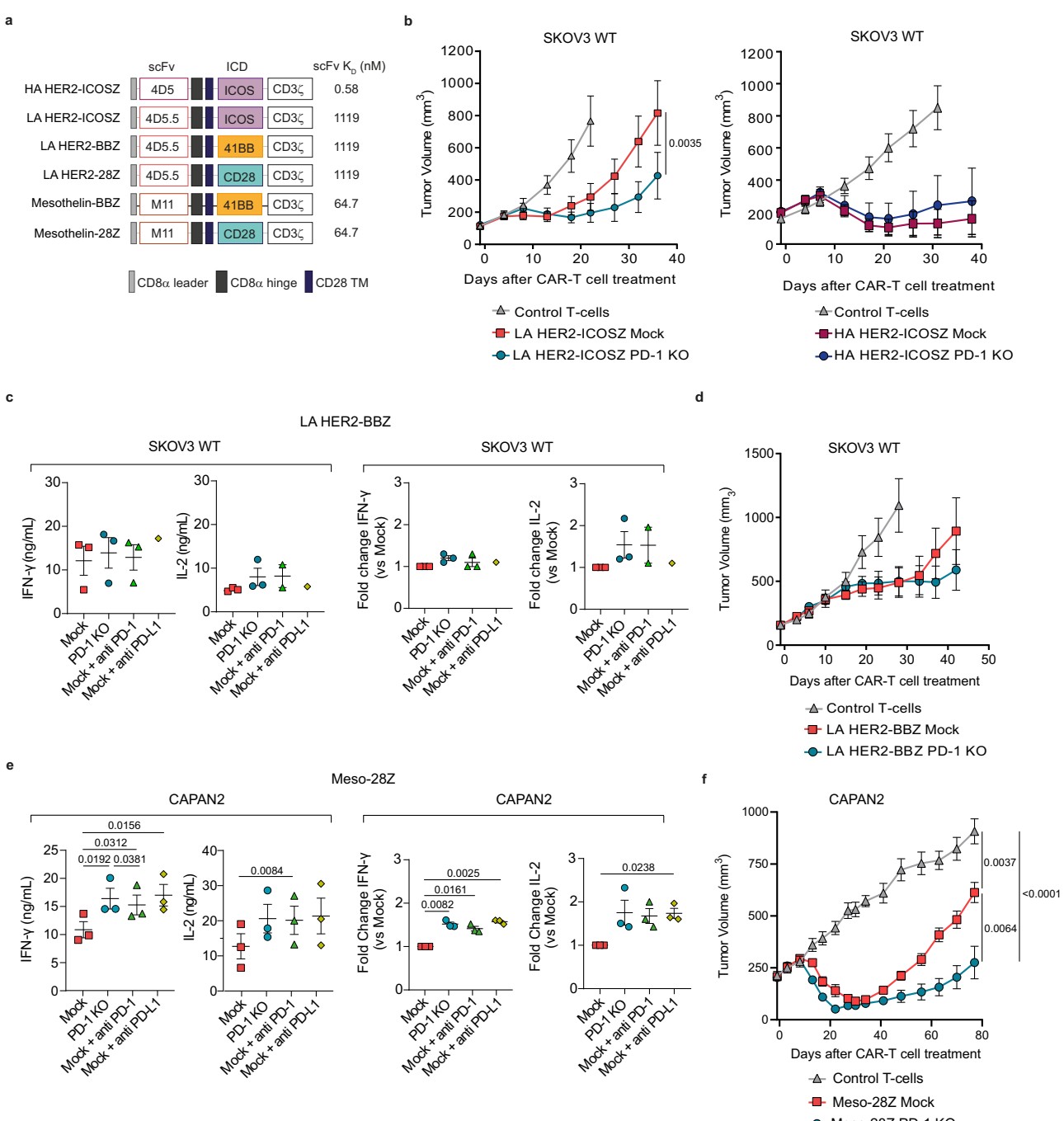

**Fig. 8 | Influence of scFv and co-stimulatory domains in PD-1/PD-L1-mediated inhibition of CAR-T cells. a** Schematic overview of CAR constructs used with their corresponding affinity values. **b** Tumor measurements of NSG mice bearing SKOV3 tumors treated with $5 \times 10^6$ control T-cells, mock or PD-1 KO LA HER2-ICOSZ CAR$^+$-T cells ($n = 10$, left panel) or with $3 \times 10^6$ control T-cells, mock or PD-1 KO HA HER2-ICOSZ CAR$^+$-T cells ($n = 7$ for control and $n = 8$ for mock and PD-1 KO, right panel). **c** Quantification of IFN-γ and IL-2 production by PD-1 KO or mock LA HER2-BBZ CAR-T cells, alone or in combination with anti PD-1 or anti PD-L1 antibodies, after 24 h of co-culture with SKOV3 tumor cells (E:T = 3:1) as measured by ELISA ($n = 3$ for all groups except for mock+anti PD-L1 with $n = 1$). **d** Tumor measurements of NSG mice bearing SKOV3 tumors treated with $3 \times 10^6$ control-T cells ($n = 8$), mock ($n = 8$) or PD-1 KO ($n = 10$) LA HER2-BBZ CAR$^+$-T cells. **e** Quantification of IFN-γ and IL-2 production by PD-1 KO or mock Meso−28Z CAR-T cells, alone or in combination with anti PD-1 or anti PD-L1 antibodies, after 24 h of co-culture with CAPAN2 tumor cells (E:T = 3:1) as measured by ELISA ($n = 3$). **f** Tumor measurements of NSG mice bearing CAPAN2 tumors and treated with $2 \times 10^6$ control T cells, mock or PD-1 KO Meso-28Z CAR$^+$-T cells ($n = 12$). Data in (**b, d, f**) are represented as mean tumor volume ± SEM and n indicates tumors per group. Data in (**c, e**) are represented as mean ± SEM of absolute levels (left panel) or fold change of indicated groups as compared to mock CAR-T cells (right panel). Data is pooled from independent experiments where each dot indicates a different donor for CAR-T generation (n). *p* values by two-way ANOVA with Sidak multiple testing correction (**b**), two-way ANOVA with Tukey's multiple testing correction (**f**), one-way ANOVA with Tukey post hoc test (**e**, absolute levels) or one-sample *T*-test (**e**, fold change) are indicated. Source data are provided as a Source Data file.

previously described[65]. Briefly, CD4+ and CD8+ T cells were negatively isolated using RosetteSep Kits (Stem Cell Technologies) and stimulated separately with CD3/CD28-activating Dynabeads (Invitrogen) at a 2:1 bead-to-cell ratio in the presence of human IL-7 and IL-15 (Miltenyi biotec) at a concentration of 10 ng/mL. Approximately 24 h after activation, T cells were transduced with CAR-encoding lentiviral vectors. Beads were removed from cultures at day 4 and T cells were counted and maintained at a concentration of $0.8 \times 10^6$ cells/mL in RPMI-1640 (Gibco) supplemented with 10% FBS (Sigma, Lot#F4531), Penicillin-Streptomycin (#15070063, ThermoFisher), 10 mM Gluta-Max (#35050061, ThermoFisher), 10 mM HEPES (15630080) and 10 ng/mL of human IL-7 and IL-15 (Milenyi Biotec). For CRISPR, CAR-T cells were electroporated with buffer alone (Mock CAR-T cells) or Cas9 and a chemically synthesized sgRNA targeting *PDCD1* exon 1 (sequence: CGACUGGCCAGGGCGCCUGU) at day 4 post-activation. Ribonucleoprotein complex was mixed at sgRNA:Cas9 molar ratio of 3.3:1, incubated during 5–20 min and returned to the incubator. CAR-T cells were then expanded ex vivo until day 10–11, when CD4+ and CD8+ T cells were mixed at a 1:1 ratio and cryopreserved. To confirm editing events in the PDCD1 locus, DNA from PD-1 KO edited CAR-T cells was extracted using the DNeasy Blood&Tissue kit (Qiagen) according to the manufacturer's protocol. The region surrounding the site of interest was amplified using the primers (forward: TTTCCCTTCCGCTCACCTCC and reverse: CAAA-GAGGGGACTTGGGCCA) and KO efficiency was assessed by Sanger sequencing and quantified by using ICE v3.0 software (Synthego) on day 10 of T-cell expansion.

## In vitro co-culture experiments

Tumor cells $(1 \times 10^5)$ were seeded in 48-well plates. Primary healthy cells $(1 \times 10^4)$ were seeded in 96-well plates. After overnight incubation, T cells were added at an effector/target ratio of 3:1. At indicated experiments, anti PD-L1 (Durvalumab) and anti PD-1 (Nivolumab) antibodies were added to CAR-T cells at a final concentration of 10 µg/mL. For cytokine secretion, supernatants were collected 24 h after co-culture, and IFN-γ and IL-2 were analyzed using the DuoSet® ELISA Development Kit (R&D Systems, DY285B/DY202) as per the manufacturer's protocol. Absorbance data was collected using Gen5 2.07 (Biotek) or iControl 2.0 (LifeSciences) software. For T-cell proliferation assays, absolute numbers of live cells were calculated for each group using trypan blue exclusion before coculture and after 6 days of incubation with tumor cells. For the experiments with the SLBs, CAR-T cells $(3 \times 10^4)$ were resuspended in 100 µL of imaging buffer, seeded onto SLBs and incubated at 37 °C for 15 min. Following incubation, 450 µL of RPMI 1640 medium supplemented with 25 mM HEPES, 10% FBS, 100 µ/mL of penicillin/streptomycin, 2 mM L-glutamine and 50 µM of 2-mercaptoethanol was added, and the cells were further incubated for 24 or 72 h, as indicated. The supernatant was collected and stored in 100 µL aliquots at −80 °C until further use. The secretion of IFN-γ was measured by performing ELISA using a commercially available kit (ELISA MAX™ Deluxe Set, BioLegend). For intracellular staining assays, tumor cells $(5 \times 10^5)$ were seeded in 12-well plates. After overnight incubation, T cells were added (effector/target ratio of 1:3). 24 h later, GolgiPlugTM (ref. 555029, BD Bioscience) was added to each well. Cell stimulation cocktail (ref. 00–4970, eBioscience) was added to the corresponding positive control wells. 4 h later, flow cytometry staining was performed as described below. For CD107a degranulation assays, target cells $(1 \times 10^5)$ were seeded in 48-well plates. After overnight incubation, T cells were added (effector/target ratio of 1:1) and incubated for 2 h at 37 °C. To enable the detection of the CD107a marker, a protein transport inhibitor containing brefeldin A, Golgi-PlugTM (ref. 555029, BD Bioscience) was added to each well along with anti-CD107a antibody. The co-culture was extended for an additional 4 h. Then, T cell staining was analyzed by flow cytometry analysis as described below.

## Mouse xenograft study

NOD/SCID/IL2-receptor γ chain knockout (NSG) mice were purchased from Jackson Laboratory. Mice were bred and maintained within the Animal Facility at the University of Barcelona, with a 12 h light/dark cycle, a temperature range of 20–24 °C and a humidity range of 45–65%. Mice health status was regularly monitored by qualified personnel. 6–8 week old female (SKOV3 and HCC1954) or male (CAPAN2) NSG mice were implanted subcutaneously with $4–5 \times 10^6$ tumor cells in a 50% solution of Matrigel (Corning) in PBS and treated intravenously with $2–5 \times 10^6$ control T cells, CAR-T cells or PD-1 KO CAR+-T cells in 100 µL of PBS when tumors reached 150–250 mm³, following previously published protocols[66]. PD-L1–blocking antibody (Durvalumab) was administered intraperitoneally at a dose of 10 mg/kg every 5 days during the specified experiment. Tumor dimensions were measured weekly with a digital calliper and volumes were calculated using the formula V = 6 x (L x W2) / π, where L is length and W is width of the tumor. Mice were sacrificed when tumors reached 1500 mm3. In some cases, this limit has been exceeded, but we ensured that that no mice remained with tumor volumes above this threshold for longer than 5 days and animals exhibiting signs of pain, discomfort, or distress were euthanized immediately.

## Immunohistochemistry staining

Tumors were harvested at the experimental endpoint and embedded in paraffin. Immunohistochemistry stainings were performed by the Biobanc HCP-IDIBAPS Core according to standard protocols. Briefly, tumor sections were incubated with a 1:100 dilution of anti-PD-L1 antibody (#15165, Cell Signaling) followed by a rabbit specific IHC polymer detection kit HRP/DAB. Slides were counterstained with hematoxylin, dehydrated and mounted. Images were obtained using a Nikon Eclipse E600 inverted microscope and a Olympus DP72 camera.

## IsoLight polyfunctionality assay

Co-cultures of CAR-T cells and SKOV3 tumor cells were established at a E:T ratio of 1:3. 20 h later, cells were collected and HER2+ tumor cells were depleted by using Anti-ErbB-2 MicroBeads (Miltenyi Biotec) following manufacturer's instructions. Enriched T cells were then stained with a cell membrane dye and an anti-CD8 AF647 antibody for differentiation of CD4+ and CD8+ T-cell populations. Subsequently, 30.000 viable cells were loaded onto the 32-plex human IsoCode Single-Cell Adaptive Immune chip (IsoPlexis) and chips were loaded into the Iso-Light machine. Data was collected and analyzed by using IsoSpeak 2.9.0 software (IsoPlexis, Branford, CT).

## Gene expression analysis

Gene expression analysis was performed using the CAR-T cell characterization panel from Nanostring Technologies (Seattle, WA). Briefly, CAR-T cells were co-cultured with SKOV3 tumor cells (effector/target ratio of 1:3) for 48 h. CD45+ cells were then flow sorted and total RNA was extracted using the RNeasy Mini kit (Qiagen). Samples were prepared according to the manufacturer's protocols for the nCounter CAR-T Characterization Panel. Cartridges were run on the nCounter SPRINT Profiler. Gene expression levels were normalized against the housekeeping genes and data analysis was conducted using the Rosalind Platform (www.rosalind.bio/nanostring). Enrichr online software (https://maayanlab.cloud/Enrichr/) was used for the analysis of biological pathways and Gene Ontology (GO) terms associated with the differentially expressed genes by using the list of under- and over-expressed genes as input.

## Flow cytometry (Surface and Intracellular stainings)

Cell viability was determined using L/D eFluor™ 450 (eBioscience, 65-0863-14) followed by surface antibody staining in FACS buffer. Cells were incubated with surface antibodies for 30 min in the dark. To detect CAR expression, cells were stained using goat anti-mouse IgG-

biotin (Jackson ImmunoResearch) followed by streptavidin-PE or streptavidin- eFluor450 (ThermoFisher, 12-4317-87 and 48-4317-82). Intracellular staining was performed with the Foxp3/Transcription Factor Staining Buffer set (ThermoFisher, 00-5523-00) according to the manufacturer's instructions. All experiments were performed on a FacsCanto 3 L, Fortessa 4 L HT and Fortessa 5 L (BD Biosciences) and the data was analyzed with FlowJo software (V.10, TreeStar). Antibodies listed on Supplementary Table 2 were used.

## Statistical analysis

All statistical analyses were performed using GraphPad Prism v9.4.1 (GraphPad Software Inc.). For comparisons of two groups, two-tailed $t$ tests or one-sample $t$ test were used. One-way analysis of variance (ANOVA) with Tukey post hoc test was used for the comparison of three or more groups in a single condition and two-way ANOVA test with Sidak or Tukey's multiple testing correction. Exact $p$ values are indicated in the figures.

## Reporting summary

Further information on research design is available in the Nature Portfolio Reporting Summary linked to this article.

## Data availability

Transcriptomics data have been deposited in the Gene Expression Omnibus database under accession number GSE252036. The remaining data are available within the Article, Supplementary Information or Source Data file. Source data are provided with this paper.

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

## Acknowledgements

This work received funding from the Spanish Ministry of Science and Innovation under a Ramon y Cajal grant (RYC2018-024442-I to S.G.) and Retos Investigación (PID2019-109546RA-I00-PI), the Innovative Medicines Initiative 2 Joint Undertaking under grant agreement No 116026 (this Joint Undertaking receives support from the European Union's Horizon 2020 research and innovation program and EFPIA),"la Caixa" Foundation under the grant agreement LCF/PR/SP23/52950004, the Spanish Association Against Cancer (LABAE20022GUED to S.G., INVES222988RODR to A.R-G. and INVES21943BRAS to F.B-M.) and a nCounter Immunology grant from Diagnóstica Longwood S.L, Nanostring. L.A. is recipient of a Rio Hortega 2020 Contract (Ministry of Health, Spain). A.P. received funding from Fundación CRIS contra el cáncer PR_EX_2021-14, Agència de Gestó d'Ajuts Universitaris i de Recerca 2021 SGR 01156, Fundación Fero BECA ONCOXXI21, Instituto de Salud Carlos III PI22/01017, Asociación Cáncer de Mama Metastásico IV Premios M. Chiara Giorgetti, Breast Cancer Research Foundation BCRF-23-198, and RESCUER, funded by European Union's Horizon 2020 Research and Innovation Programme under Grant Agreement No. 847912. We are indebted to the Biobank core facility, to the Flow Cytometry and Cell Sorting core facility of Institut d'Investigacions Biomèdiques August Pi i Sunyer (IDIBAPS) for their technical help and to the animal facility of the Universitat de Barcelona. We thank S. Bragado from IsoPlexis and D. Benítez from the Immunology department for advice and assistance with the single-cell secretome assay and P. Galván, O. Castillo and P. Blasco from the Translational genomics and targeted therapies in solid tumors group for assistance with the gene expression assay. We thank A. Gros, R. Alemany and M. Juan for helpful scientific discussions. Panels in Fig. 3a and Fig. 7h were created with BioRender.com, released under a Creative Commons Attribution-NonCommercial-NoDerivs 4.0 International license, or with Adobe Illustrator, respectively, as indicated in figure legends.

## Author contributions

I.A.-S. and A.R-G. designed and performed the experiments, analyzed and interpreted the data and wrote the manuscript. M.G.-A., V.M., S.C., L.A. and M.N. performed experiments. J.C. performed in vivo experiments. F.B-M assisted with transcriptomic data analysis. H.C. developed some of the CAR constructs used in the study and assisted with in vivo experiments. B.M. developed some of the CAR constructs and tumor cell lines used in the study. M.S-C helped with lentiviral production. P.B.

assisted with in vivo experiments. J.B.H. designed experiments, provided conceptual guidance and revised the manuscript. A.P. and A.U-I provided conceptual guidance and revised the manuscript. S.G supervised the project, including design of experiments, data analysis and interpretation and manuscript writing.

## Competing interests

S.G. is an inventor on patents related to CAR-T cell therapy, filed by the University of Pennsylvania and licensed to Novartis and Tmunity, and has received commercial research funding from Gilead. A.R-G. is an inventor on patents related to CAR-T cell therapy, filed by the University of Pennsylvania. The remaining authors declare no competing interests.
