## [Peer Review File · Nature Communications]

CAR affinity modulates the sensitivity of CAR-T cells to PD-1/
PD-L1-mediated inhibitionReviewers' Comments:

Reviewer #1:

Remarks to the Author:

The study by Andreu-Saumell et al. investigates the relationship between the affinity of a T-cell chimeric antigen receptor (CAR) and its response to the PD1:PD-L1 axis by using in vitro and preclinical in vivo models. They conclude that high-affinity CAR T cells are generally unresponsive to PD1:PD-L1 (not suppressed by PD-1, but also not exhausted by PD-1 knock-out), whereas low-affinity CAR T cells benefit from the PD1:PD-L1 blockade. Moreover, they propose that the role of PD-1 is different in CAR T cells with different signaling domains.

The experiments are well-planned, described, and interpreted. The manuscript provides interesting findings in the field of CAR immunotherapy. The role of PD-1 in CAR therapy is a long-standing and controversial topic (as also stated by the authors). However, the relatively minimalistic scope of this work (e.g., largely limited to a single tumor line - SKOV3 ovarian cancer and one CAR with low and one CAR with high affinity) would not be able to resolve this question completely, although it brings an interesting and conclusive insight. I appreciate the use of modified SKOV3 lines with varied expression of PD-L1, which adds a high level of credibility to the work. The use of M11 CAR and the CAPAN2 lines also increases relevance.

I have several relatively minor comments that should be addressed during eventual revisions.

1. Concerning the transcriptomics experiments - it would be great to show PCA of all the analyzed samples, which would be more informative on the overall similarities/differences between samples/conditions than the paired heatmaps using a set of selected genes shown here.
2. I am not sure about the Nat. Com. editorial policy, but it is a standard in the field that the transcriptomics data are deposited in a public repository, which seems not to be the case here.
3. In Figure 3C, the authors compare normalized IFNG production using a paired T test. However, this type of test cannot be used for such data, where one of the datasets is arbitrarily set as 1 (here Mock + Her2), as it assumes normal distribution of the data. The same is in Fig. 6E (right) and in Supp. Fig. 2,
4. I would appreciate a more complete description of all used CARs in Fig. 6A. The CD28-based 4D5 and 4D5.5 CARs used in most experiments of the study are not shown. Also M11-BBZ was used in Supp. Figure 6 and is missing in Fig. 6A. As the 4-1BB containing CAR is called HER2-BBZ in other panels and in the text - this should be noted in Fig. 6A as well. Moreover, showing that 4D5 is HA, whereas M11 and 4D5.5 is LA here would help.

Reviewer #2:

Remarks to the Author:

In this paper, Andreu-Saumell and Rodriguez-Garcia et al. demonstrate that CAR affinity affects sensitivity to PD-1-mediated inhibition. There is conflicting literature in the field on whether PD-1 axis disruption improves, impairs, or has no effect on CAR-T cell therapy, and it is unclear what factors may dictate response of CAR-T cells to PD-1 blockade. This study focuses on defining how the variable of CAR affinity influences sensitivity to PD-1-mediated inhibition. Using tumor cell lines and lipid bilayers engineered to express variable levels of PD-L1, the authors show that PD-L1 inhibits the effector function and proliferation of low-affinity CAR-T cells but has little to no effect on high-affinity CAR-T cells. This inhibition of low affinity CARs can be reversed by PD-1 knockout or PD-1/PD-L1 blockade. The authors also show that these differences hold true for CARs carrying CD28 or ICOS costimulatory domains, but not for CARs carrying 4-1BB costimulatory domains.

Overall, the focus on the variable of CAR affinity as a factor influencing sensitivity to PD-1 inhibition is novel, and the experiments are carefully controlled. However, additional data are needed to support the claim that CAR affinity is a major driver of resistance to PD-1 inhibition. In particular, it is important to show that differences in CAR surface expression between low and high affinity CARs is not a confounding factor, and to extend analyses to another pair of low/high affinity CARs to determine if this is a generalizable claim. Areas for improvement are detailed below:

Major comments:

- 1) Mutations to tune affinity of scFvs can often also affect surface CAR expression or receptor clustering on the cell surface, which could impact downstream signaling and activation independently of receptor affinity. Although the authors show surface CAR expression as a percentage in Fig. 1D, they should also show the median fluorescence intensity (MFI) of CAR staining among the CAR+ cells and show representative histograms of the staining. It is possible that though the percentage of CARs is similar between LA and HA CARs, the HA CARs may show higher MFI indicating higher surface expression. If so, this would confound interpretation of the paper's results, as it is unclear whether the differences between the two CARs are due to affinity or to surface level expression.
- 2) It would also help to show surface CAR expression for the Her2 CARs with CD28, ICOS, and 4-1BB signaling domains in Fig.6, as costim domain has been shown to affect CAR surface expression. This is important to understand whether differences with different costim domains are due to costimulatory signaling or due to differences in receptor expression.
- 3) To make the general claim that "CAR affinity modulates the sensitivity of CAR-T cells to PD-1/PD-L1-mediated inhibition", it is important to show similar differences with at least one other pair of scFvs targeting the same antigen but differing in affinity. The authors show that another low affinity scFv targeting mesothelin is also sensitive to PD-1 inhibition in Fig. 6E/F; however, it is not clear if this is related to receptor affinity or some other property of the scFv/target antigen unless a high-affinity mesothelin scFv is also tested by comparison.
- 4) Does the resistance of high-affinity CARs resistant to PD-1 inhibition depend on CAR antigen density? That is, do high-affinity CARs become susceptible to PD-1 inhibition at lower antigen densities. A strength of the lipid bilayer model system is the ability to titrate both the CAR antigen density as well as PD-L1 levels, but the effects of CAR antigen density are not shown. This is an important question to answer, as the authors argue that high affinity CARs could be used clinically to overcome PD-1 inhibition, but the utility of this strategy may depend on target antigen density.

Minor comments:

- 5) Fig. 1H/J: It looks like mock T cells make less IFN γ in the presence of PDL1-high tumors relative to PDL1 KO tumors, but this is not reversed by PD-1/PD-L1 blockade or PD-1 KO. Can the authors comment on why this is? Is Her2 expression on the different tumor cell lines similar?
- 6) Fig.5G needs replicates and statistical testing.

Reviewer #3:

Remarks to the Author:

The work from Andreu-Saumell et al present clear and convincing data discussing the impact of different parameters, especially the affinity of the scFv region, in determining the sensitivity of CAR-T cells to PD-1/PD-L1 axis inhibition. In particular, the authors aim at clarifying conflicting results present in literature on this regard through the use of robust preclinical models for systematic interrogation of different CAR-T cell products. Noteworthy, they showed that both pharmacological and genetic disruption of the PD1-1/PD-L1 axis on CAR-T cells can enhance anti-tumor activity only when paired with a low-affinity (LA) CAR construct, while high-affinity (HA) CAR-T cell products do not show any benefit. The finding is supported by several in vitro and in vivo assays, as well as transcriptomic and single cell secretome analysis of LA vs HA CAR-T cells. The positive effect of PD1/PD-L1 disruption on LA-CARs has been explored using an anti-HER2 CAR and confirmed with an anti-mesothelin CAR. Finally, it was shown that this concept is valid for CARs carrying CD28 and ICOS, but not with

constructs including 41BB.

Overall, the research article comes across as clear and well-thought. Language employed is great and the logical flow is both sound and easy to follow. Assays are performed with a suitable number of replicates, and in particular in vivo work is very convincing. The idea of CAR affinity being a key determinant for the sensitivity to anti PD-1 treatment is fascinating and well executed. Albeit the topic is very specific, I can surely see the potential impact of this research article in the field.

Main points

The authors did a thorough job at demonstrating that only LA CAR-T cells benefit from PD-1 genetic ablation, but I would advise to deepen their argumentations on the advantages of using LA CAR-T cells + PD1 blockade versus HA CAR-T cells, which are similar in terms of antitumor activity. The topic is only briefly mentioned in discussion. For example, I recommend discussing why HA CAR-T cells are expected to be less safe, mentioning the impact on different types of toxicity. Similarly, I suggest to better discuss why HA CAR-T cells could pose risk in terms of increased T-cell exhaustion. Related to this, it could be relevant to include supporting data from in vivo experiments analyzing the expansion/exhaustion profile of circulating CAR-T cell products and serum cytokines in the different conditions.

I would appreciate if the authors could expand their setting in the lipid bilayer system to take into account the effect of varying target antigen density.

In Figure 5 did the authors expect such a low percentage of polyfunctional cells (less than 5%) in LA CAR-T cell products even when combined with PD1 ablation? It seems very low even compared to the 40% of polyfunctional cells present in HA CAR-T cell products, which again raises the question on the comparison of LA CAR+ PD1 ablation vs HA CAR.

I would recommend moving to main figure all data concerning co-treatment with anti PDL-1 blocking antibodies, since it brings further support to the authors claim, and expanding on the advantages of a co-treatment vs genetic ablation in discussion.

Minor points

Did the authors expect and have an explanation as to why in Figure 2 PD1-KO CAR-T cells are less efficient than mock cells in controlling the growth of PDL1-KO tumors?

In Fig S1b Capan2 cell line seems bi-modal rather than low-expressing.

REVIEWER COMMENTS

Reviewer #1 (Remarks to the Author):

The study by Andreu-Saumell et al. investigates the relationship between the affinity of a T-cell chimeric antigen receptor (CAR) and its response to the PD1:PD-L1 axis by using in vitro and preclinical in vivo models. They conclude that high-affinity CAR T cells are generally unresponsive to PD1:PD-L1 (not suppressed by PD-1, but also not exhausted by PD-1 knock-out), whereas low-affinity CAR T cells benefit from the PD1:PD-L1 blockade. Moreover, they propose that the role of PD-1 is different in CAR T cells with different signaling domains.

The experiments are well-planned, described, and interpreted. The manuscript provides interesting findings in the field of CAR immunotherapy. The role of PD-1 in CAR therapy is a long-standing and controversial topic (as also stated by the authors). However, the relatively minimalistic scope of this work (e.g., largely limited to a single tumor line - SKOV3 ovarian cancer and one CAR with low and one CAR with high affinity) would not be able to resolve this question completely, although it brings an interesting and conclusive insight. I appreciate the use of modified SKOV3 lines with varied expression of PD-L1, which adds a high level of credibility to the work. The use of M11 CAR and the CAPAN2 lines also increases relevance.

I have several relatively minor comments that should be addressed during eventual revisions.

1. Concerning the transcriptomics experiments - it would be great to show PCA of all the analyzed samples, which would be more informative on the overall similarities/differences between samples/conditions than the paired heatmaps using a set of selected genes shown here.

As suggested by the reviewer, we represented PCA plots of the analyzed samples. However, the panel of analyzed genes is only composed of 780 genes and many of them are involved in antigen presentation machinery pathways as well as TCR diversity and therefore there may be donor-dependent. For those reasons, when we performed PCA analysis, samples tend to group by healthy donor rather than by PD-1 phenotype (mock or PD-1 KO).

In an attempt to correct for this bias, we have performed PCA analysis in which genes corresponding to TCR diversity and antigen processing and presentation categories had been excluded. As observed in the graphs below, while samples from high affinity (HA) HER2-28Z CAR-T cells grouped by normal donor (ND, lower panels), samples from low affinity (LA) HER2-28Z CAR-T cells seem to group more by PD-1 phenotype (upper panels). We believe that the conclusions obtained by PCA analysis of this Nanostring panel, while in line with our message, is not solid enough to be included in the main manuscript.

Principal Component Analysis

2. I am not sure about the Nat. Com. editorial policy, but it is a standard in the field that the transcriptomics data are deposited in a public repository, which seems not to be the case here.

As suggested by the reviewer, we have now deposited the transcriptomics data corresponding to the nanostring experiment in the Gene Expression Omnibus database under accession number GSE252036. The token for the reviewers is ujvgsqkhvittel. We have also included GEO accession number in the Data Availability section.

3. In Figure 3C, the authors compare normalized IFNG production using a paired T test. However, this type of test cannot be used for such data, where one of the datasets is arbitrarily set as 1 (here Mock + Her2), as it assumes normal distribution of the data. The same is in Fig. 6E (right) and in Supp. Fig. 2,

We thank the reviewer for the comment. We have now used an appropriate statistical test taking into consideration the fact that one dataset is arbitrarily set as 1 (one sample t test) and updated the corresponding figures, accordingly.

4. I would appreciate a more complete description of all used CARs in Fig. 6A. The CD28-based 4D5 and 4D5.5 CARs used in most experiments of the study are not shown. Also M11-BBZ was used in Supp. Figure 6 and is missing in Fig. 6A. As the 4-1BB containing CAR is called HER2-BBZ in other panels and in the text - this should be noted in Fig. 6A as well. Moreover, showing that 4D5 is HA, whereas M11 and 4D5.5 is LA here would help.

As the reviewer suggest, we have now updated Fig. 8a (previously Fig. 6a) and Supplementary Fig. 10d. to include a more complete description of all the CAR constructs used in the studies depicted in Fig. 8 and Supplementary Fig. 10.

Reviewer #2 (Remarks to the Author):

In this paper, Andreu-Saumell and Rodriguez-Garcia et al. demonstrate that CAR affinity affects sensitivity to PD-1-mediated inhibition. There is conflicting literature in the field on whether PD-1 axis disruption improves, impairs, or has no effect on CAR-T cell therapy, and it is unclear what factors may dictate response of CAR-T cells to PD-1 blockade. This study focuses on defining how the variable of CAR affinity influences sensitivity to PD-1-mediated inhibition. Using tumor cell lines and lipid bilayers engineered to express variable levels of PD-L1, the authors show that PD-L1 inhibits the effector function and proliferation of low-affinity CAR-T cells but has little to no effect on high-affinity CAR-T cells. This inhibition of low affinity CARs can be reversed by PD-1 knockout or PD-1/PD-L1 blockade. The authors also show that these differences hold true for CARs carrying CD28 or ICOS costimulatory domains, but not for CARs carrying 4-1BB costimulatory domains.

Overall, the focus on the variable of CAR affinity as a factor influencing sensitivity to PD-1 inhibition is novel, and the experiments are carefully controlled. However, additional data are needed to support the claim that CAR affinity is a major driver of resistance to PD-1 inhibition. In particular, it is important to show that differences in CAR surface expression between low and high affinity CARs is not a confounding factor, and to extend analyses to another pair of low/high affinity CARs to determine if this is a generalizable claim. Areas for improvement are detailed below:

Major comments:

1) Mutations to tune affinity of scFvs can often also affect surface CAR expression or receptor clustering on the cell surface, which could impact downstream signaling and activation independently of receptor affinity. Although the authors show surface CAR expression as a percentage in Fig. 1D, they should also show the median fluorescence intensity (MFI) of CAR staining among the CAR+ cells and show representative histograms of the staining. It is possible that though the percentage of CARs is similar between LA and HA CARs, the HA CARs may show higher MFI indicating higher surface expression. If so, this would confound interpretation of the paper's results, as it is unclear whether the differences between the two CARs are due to affinity or to surface level expression.

As suggested by the reviewer, we have now included a representative histogram showing expression of LA and HA HER2-28Z CAR as **Fig. 1d** as well as a quantification of the MFI in three healthy donors as **Supplementary Fig. 1c**. We show that, although differences in MFI quantification are statistically significant probably due to small standard deviations, expression levels of both CARs are very similar, with only 1.14-fold change difference. In this case, we do not believe that differential CAR expression levels are the reason accounting for differences observed between LA and HA HER2-28Z in terms of sensitivity to PD-L1-mediated inhibition.

Having that in mind, we also agree with the reviewer that CAR levels might also play a role in determining resistance to PD-1/PD-L1 axis, as we had hypothesized briefly in the discussion section of the original manuscript. We have now more deeply explored the role of CAR expression on sensitivity to PD-1/PD-L1 axis. We have found that, when conducting *in vitro* experiments in which we co-cultured SKOV3 PD-L1 High tumor cells and CAR-T cell products with high frequencies of CAR-positive cells (75-90%), the advantage provided by PD-1 KO in LA HER2-28Z CAR-T products with typical CAR frequencies (50-65%) was lost, to a similar extent of what we observed with the HA CARs.

These results highlight that, although CAR affinity has a pivotal role as a determinant of CAR-T cell sensitivity to PD-1/PD-L1 axis, there are other factors implicated, including the levels of CAR expression.

We have now incorporated these results as **Fig. 7h-j**, and included the following text in the results new section entitled **“Target antigen densities and CAR expression play a role in determining sensitivity to PD-L1”**: *“Based on these results, we hypothesized that PD-L1-mediated inhibition could potentially be overcome at a certain threshold of T-cell activation, and that this could also be achieved by utilizing T-cell products with high CAR expression. To validate this hypothesis, we conducted studies with T-cell products containing more than 75% CAR+ T cells (Fig. 7h). In this scenario, the advantage provided by PD-1 KO in the LA CAR-T cells was lessened (Fig. 7i), similar to our observations in the HA setting (Fig. 7j). Overall, our results demonstrate that although CAR affinity is pivotal in determining sensitivity of CAR-T cells to PD-1/PD-L1 axis, other factors such as antigen density and CAR expression levels may also play a role.”*

2) It would also help to show surface CAR expression for the Her2 CARs with CD28, ICOS, and 4-1BB signaling domains in Fig.6, as costim domain has been shown to affect CAR surface expression. This is important to understand whether differences with different costim domains are due to costimulatory signaling or due to differences in receptor expression.

As suggested by the reviewer, we have now included surface CAR expression data for the HER2 CARs with CD28, ICOS and 4-1BB co-stimulation domain in terms of percentage of CAR⁺-T cells and MFI as **Supplementary Fig. 10a-c**. We observed that ICOS-based CARs were expressed at significantly lower frequencies and MFI as compared to CARs containing CD28, but still, both of them are sensitive to PD-L1-mediated inhibition. By contrast, 4-1BB co-stimulated CARs were expressed at similar levels as compared to CD28-based CARs and showed increased resistance to PD-L1. Based on these results, we conclude that differential sensitivities of CARs featuring distinct co-stimulatory domains to PD-1/PD-L1 axis are more likely related to differences in PD-1 expression levels as well as to signaling pathways intrinsic to each co-stimulation domain. In support of this, previously published works already reported a higher resistance of 4-1BB-based CARs to PD-1/PD-L1 axis as compared to those with CD28 as co-stimulatory domain (Zolov, S.N. et al. Programmed cell death protein 1 activation preferentially inhibits CD28.CAR-T cells. *Cytotherapy*, 2018. 20(10): p. 1259-1266; Cherkassky, L., et al., Human CAR T cells with cell-intrinsic PD-1 checkpoint blockade resist tumor-mediated inhibition. *J Clin Invest*, 2016. 126(8): p. 3130-44).

We have now modified the manuscript to include the following statement in the results section: *“Differential CAR expression was ruled out as a potential reason for the differing sensitivity to PD-L1-mediated inhibition among constructs with distinct co-stimulatory domains, as ICOS-based CARs, despite being expressed at lower levels as compared to CD28, were still sensitive to PD-1/PD-L1 axis. In contrast, 4-1BB-based CARs exhibited comparable expression levels to CD28 but demonstrated greater resistance to inhibition by PD-L1 (Supplementary Fig. 10a-10c).”*

3) To make the general claim that CAR affinity modulates the sensitivity of CAR-T cells to PD-1/PD-L1-mediated inhibition, it is important to show similar differences with at least one other pair of scFvs targeting the same antigen but differing in affinity. The authors show that another low affinity scFv targeting mesothelin is also sensitive to PD-1 inhibition in Fig. 6E/F; however, it is not clear if this is related to receptor affinity or some other property of the scFv/target antigen unless a high-affinity mesothelin scFv is also tested by comparison.

As suggested by the reviewer, during the course of the revision of this manuscript we have extended our analyses to include another pair of CARs targeting the same antigen with different affinities.

For that, we generated CAR-T cells targeting folate receptor beta (FR β) with either high (Kd=2.48nM) or low/intermediate affinity (Kd=54.3nM) (Lynn RC, Powell DJ Jr. et al. High-affinity FR β -specific CAR T cells

eradicate AML and normal myeloid lineage without HSC toxicity. *Leukemia*. 2016 Jun;30(6):1355-64. doi: 10.1038/leu.2016.35. Epub 2016 Feb 22. PMID: 26898190; PMCID: PMC4889499). In parallel, we genetically engineered our SKOV3 PD-L1 KO and SKOV3 PD-L1 high tumor cells to overexpress the CAR antigen, FR β (as FR β is not typically expressed in tumor cells but in M2 macrophages in the context of solid tumors).

We then measured IFN- γ secretion after 24 hours of co-culture of LA or HA FR β -28Z CAR-T cells expanded from four different healthy donors and the target cancer cells. Similar levels of IFN- γ were released by mock and PD-1 KO CAR-T cells in the absence of PD-L1 expressed by tumor cells by both LA and HA CAR-T cells. By contrast, significantly higher levels of IFN- γ were secreted by LA PD-1 KO as compared to mock FR β -28Z CAR-T cells in co-culture with high PD-L1-expressing tumor cells. These differences between mock and PD-1 KO, however, were not seen for HA FR β -28Z CAR-T cells, which released similar levels of IFN- γ regardless of PD-L1 presence. These results are aligned with our observations with HER2-28Z CAR-T cells.

We have now included the new CARs schema in **Fig. 1c**, IFN- γ secretion data as **Fig. 1k** and the data related to the model as **Supplementary Fig. 3** (i.e. surface CAR and PD-1 expression on T cells, FR β and PD-L1 expression in tumor cells). We also modified the manuscript to incorporate the following statements in the results section: *"We validated these findings in an alternative pair of CARs targeting FR β with different affinities (Fig. 1c and Supplementary Fig. 3a-3c). By using the SKOV3-based PD-L1 cellular model in which we overexpressed FR β (Supplementary Fig. 3d), we found significantly higher levels of IFN- γ secreted by PD-1 KO LA CAR-T cells as compared to mock in the presence of PD-L1 (Fig. 1k, left panel). These differences were not observed in co-culture with the PD-L1 KO cell line or between PD-1 KO and mock HA CAR-T cells, which released similar levels of IFN- γ regardless of PD-L1 presence (Fig. 1k, right panel). While the differences in affinity were less pronounced for the FR β -targeting CAR pair (21.89-fold), these findings align with our observations from the HER2 model."*

4) Does the resistance of high-affinity CARs resistant to PD-1 inhibition depend on CAR antigen density? That is, do high-affinity CARs become susceptible to PD-1 inhibition at lower antigen densities. A strength of the lipid bilayer model system is the ability to titrate both the CAR antigen density as well as PD-L1 levels, but the effects of CAR antigen density are not shown. This is an important question to answer, as the authors argue that high affinity CARs could be used clinically to overcome PD-1 inhibition, but the utility of this strategy may depend on target antigen density.

Similar to what the reviewer points out, we also hypothesized that high affinity CARs may become susceptible to PD-L1-mediated inhibition at low densities of CAR antigen, as briefly mentioned in the discussion section of the original manuscript.

We have now addressed this issue by conducting new experiments in two different *in vitro* models.

First, we have used the lipid bilayer model system described in **Fig. 3a** now with titrated levels of CAR antigen as suggested by the reviewer. In lipid bilayers that contained fixed high levels of PD-L1 molecules (200ng), HER2 molecules were titrated down covering a range from 0.005 to 5ng. We then added CAR-T cells on top of these bilayers, and measured IFN- γ secretion at 72 hours (instead of 24 hours as done in the experiment depicted in **Fig.3**). In the LA HER2-28Z CAR setting, PD-L1 completely inhibited IFN- γ secretion by mock CAR-T cells, which was restored by PD-1 KO at all levels of HER2 tested, in line with previous observations. Presence of PD-L1 did not inhibited IFN- γ secretion by mock HA HER2-28Z CAR-T cells, which remained constant through all HER2 densities. Of note, at the lowest antigen levels, mock and PD-1 KO exhibited similar behavior, while as HER2 levels increased, PD-1 KO appeared to have a detrimental effect.

Second, we have used a cellular model based on a triple-negative breast cancer cell line, MDA-MB-468, engineered to express either low or high levels of HER2. As this cell line expresses low levels of PD-L1, we also modified them to express constitutive high levels of PD-L1.

By using this model, we have observed that, consistent to what we see in the SKOV3 model with high HER2 expression, when levels of HER2 are high, PD-1 KO only benefit LA CAR-T cells while does not affect HA CAR-T cells in terms of cytokine production as measured by ELISA or intracellular cytokine staining (ICS). By contrast, when co-cultured with HER2-low cells, PD-1 KO also provided an advantage to HA HER2-28Z CAR-T cells at least as measured by ICS, while differences as quantified by ELISA were minor and only significant for IL-2 but not for IFN- γ . LA HER2-28Z CAR-T cells were not reactive against the HER2-low cell line.

We have incorporated the new results related to CAR antigen density as new **Fig. 7a and 7b** (IFN- γ in lipid bilayer system), **Supplementary Fig. 9** (HER2 and PD-L1 staining in the MDA-MB-468 model), **Fig. 7c,d** and **Supplementary Fig. 9b** (IFN- γ and IL-2 in the cellular model, respectively), and **Fig. 7e-g** and **Supplementary Fig. 9 c-d** (ICS in the cellular model). We have also included the following statements in the results new section entitled ***“Target antigen densities and CAR expression play a role in determining sensitivity to PD-L1”***: *“We then investigated how target antigen densities influence the heightened resistance of HA CAR-T cells to the PD-1/PD-L1 axis. We hypothesized that HA CAR-T cells might become susceptible to this inhibitory pathway under conditions of low antigen densities. To explore this, we took advantage of the lipid bilayer model outlined in Fig. 3a to titrate down HER2 densities while maintaining constant high levels of PD-L1. In this controlled environment, HA CAR-T cells remained unaffected by PD-L1, as indicated by comparable levels of IFN- γ released by mock CAR-T cells across all HER2 conditions (Fig. 7b). Of note, at the lowest antigen levels, mock and PD-1 KO exhibited similar behavior, while as HER2 levels increased, PD-1 KO appeared to have a detrimental effect (Fig. 7b). In the LA setting, PD-1 KO conferred an advantage to CAR-T cells at all antigen density conditions tested (Fig. 7a). Next, we employed a cellular model based on a triple-negative breast cancer cell line, MDA-MB-468, engineered to express either low or high levels of HER2 along with constitutive high levels of PD-L1 (Supplementary Fig. 9a). Consistent with our observations in the SKOV3 model, PD-1 KO provided an advantage to LA CAR-T cells when HER2 levels were high. However, under conditions of high antigen densities, HA HER2-28Z CAR-T cells did not benefit from PD-1 KO (Fig. 7c and 7e-7f). Conversely, in co-culture with HER2-low cells, PD-1 KO conferred an advantage to HA CAR-T cells under certain settings, reaching statistical significance in terms of increased percentage of polyfunctional T cells producing both IFN- γ and TNF- α (Fig. 7e and 7g) and IL-2 secretion (Supplementary Fig. 9b), but not in IFN- γ as measured by ELISA (Fig. 7d). In line with toxicity results in Fig. 6, LA HER2-28Z CAR-T cells did not exhibit reactivity in low antigen conditions.”*

In addition, we have modified the Discussion section to include the following paragraph commenting on the role of both CAR expression levels and target antigen densities, both aspects highlighted by reviewer #2: *“In this work, we emphasize CAR affinity as a central factor influencing sensitivity to PD-1/PD-L1 axis. Beyond this, we also explored the implications of both target antigen densities and CAR expression frequencies. Intriguingly, we observed that the enhanced resistance of HA CAR-T cells to PD-L1 was attenuated in the presence of low levels of CAR antigen. Conversely, the use of products with higher CAR expression appeared to mitigate the sensitivity of LA CAR-T cells to PD-L1-mediated suppression. These findings lead us to postulate that inhibitory effects of PD-L1 on T cells can be overcome when T cell activation reaches a certain threshold. According to our results, this threshold of activation can be attainable not only through the utilization of a high affinity CAR but also by the presence of high CAR frequencies. However, in principle this latter strategy lacks the potential of being translated into the clinical setting due to risk of genotoxicity, given the higher*

amounts of vector required. Moreover, the use of CAR T cell products with high CAR expression has been correlated to worse clinical responses due to accelerated T cell exhaustion [39]. Interestingly, in apparent contradiction to this, one of the most successful CAR-T trials in solid tumors to date in term of response rates, utilized products with >70% CAR transduction, potentially contributing to the success [2] [40]. Overall, our results highlight the complexity of CAR-T cell activity regulation, involving numerous interplaying factors.”

Minor comments:

5) Fig. 1H/J: It looks like mock T cells make less IFN γ in the presence of PDL1-high tumors relative to PDL1 KO tumors, but this is not reversed by PD-1/PD-L1 blockade or PD-1 KO. Can the authors comment on why this is? Is Her2 expression on the different tumor cell lines similar?

As the reviewer points out, it appears that levels of IFN- γ produced by HA CAR-T cells in the presence of PD-L1 high-expressing tumor cells are lower as compared to those released in co-culture with PD-L1 KO cells. We have now confirmed by flow cytometry that HER2 expression is similar between SKOV3 PD-L1 KO and SKOV3 PD-L1 high cancer cells, and comparable to HER2 levels found in wild type cells (WT), ruling out differential HER2 expression as a potential reason explaining this observation.

The reason for which this may happen remains unknown. However, it is relevant to notice that in the lipid bilayer system, in which only HER2, PD-L1, and ICAM-1 molecules are present, levels of IFN- γ released by mock HA HER2-28Z CAR-T cells are very similar in the absence or presence of PD-L1 (**Fig. 3b**, right panel). This observation suggests that there might be alternative inhibitory pathways acting on HA HER2 CAR-T cells in the cellular model and therefore this inhibition can not be reverted by PD-1 KO.

6) Fig.5G needs replicates and statistical testing.

As suggested by the reviewer, we have now extended the intracellular cytokine staining study to include five different healthy donors. We have incorporated the new data including appropriate statistical testing as **Fig. 5h** and moved data regarding PMA-ionomycin controls to **Supplementary Fig. 7b**, for the sake of space.

Reviewer #3 (Remarks to the Author):

The work from Andreu-Saumell et al present clear and convincing data discussing the impact of different parameters, especially the affinity of the scFv region, in determining the sensitivity of CAR-T cells to PD-1/PD-L1 axis inhibition. In particular, the authors aim at clarifying conflicting results present in literature on this regard through the use of robust preclinical models for systematic interrogation of different CAR-T cell products. Noteworthy, they showed that both pharmacological and genetic disruption of the PD1-1/PD-L1 axis on CAR-T cells can enhance anti-tumor activity only when paired with a low-affinity (LA) CAR construct, while high-affinity (HA) CAR-T cell products do not show any benefit. The finding is supported by several in vitro and in vivo assays, as well as transcriptomic and single cell secretome analysis of LA vs HA CAR-T cells. The positive effect of PD1/PD-L1 disruption on LA-CARs has been explored using an anti-HER2 CAR and confirmed with an anti-mesothelin CAR. Finally, it was shown that this concept is valid for CARs carrying CD28 and ICOS, but not with constructs including 41BB.

Overall, the research article comes across as clear and well-thought. Language employed is great and the logical flow is both sound and easy to follow. Assays are performed with a suitable number of replicates, and in particular in vivo work is very convincing. The idea of CAR affinity being a key determinant for the sensitivity to anti PD-1 treatment is fascinating and well executed. Albeit the topic is very specific, I can surely see the potential impact of this research article in the field.

Main points

The authors did a thorough job at demonstrating that only LA CAR-T cells benefit from PD-1 genetic ablation, but I would advise to deepen their argumentations on the advantages of using LA CAR-T cells + PD1 blockade versus HA CAR-T cells, which are similar in terms of antitumor activity. The topic is only briefly mentioned in discussion. For example, I recommend discussing why HA CAR-T cells are expected to be less safe, mentioning the impact on different types of toxicity. Similarly, I suggest to better discuss why HA CAR-T cells could pose risk in terms of increased T-cell exhaustion. Related to this, it could be relevant to include supporting data from in vivo experiments analyzing the expansion/exhaustion profile of circulating CAR-T cell products and serum cytokines in the different conditions.

As suggested by the reviewer, we have now deepened our argumentations on why HA CAR-T cells are expected to be less safe as compared to PD-1 KO LA CAR-T cells.

It is well-established that high affinity CARs recognize lower levels of target antigen and, consequently, their ability to discriminate between tumor cells and healthy cells expressing low levels of antigen is diminished. In fact, a serious event occurred in the context of HER2-targeting CAR therapy where the use of a HA CAR (based on the scFv 4D5, as employed in our study) led to a fatal outcome in a patient with colon cancer metastatic to the lungs and liver. This was attributed to a potential CAR-mediated recognition of low levels of HER2 on lung epithelial cells (Morgan R, Rosenberg S et al. Case report of a serious adverse event following the administration of T cells transduced with a chimeric antigen receptor recognizing ERBB2. Mol Ther 2010;18:843–51). A previous work by Liu et al. (Liu X, June CH, Zhao Y et al. Affinity-Tuned ErbB2 or EGFR Chimeric Antigen Receptor T Cells Exhibit an Increased Therapeutic Index against Tumors in Mice. Cancer Res. 2015 Sep 1;75(17):3596-607. doi: 10.1158/0008-5472.CAN-15-0159. PMID: 26330166; PMCID: PMC4560113.) described several affinity-tuned scFv specific for HER2 based on trastuzumab scFv 4D5 and reported an increased therapeutic potential of low affinity CARs with reduced toxicity and maintained antitumor efficacy. In fact, the HER2 CARs used in our work as a model were described in the work by Liu et al.

In order to address the reviewer's comment, we conducted additional studies during the revision of this manuscript. More specifically, we established co-cultures of LA or HA HER2-28Z CAR-T cells with or without PD-1 KO and a panel of human healthy primary cells including Epidermal Keratinocytes (NHEK), Renal Epithelial Cells (HREpC), Pulmonary Artery Endothelial Cells (HPAEC), Pulmonary Artery Smooth Muscle Cells (HPASMC) and assessed T cell activation based on the degranulation marker CD107- α and the production of cytokines IFN- γ and IL-2. In the work by Liu, similar experiments had been conducted by using 4-1BB-based affinity-tuned HER2 CARs. In their work, low but still detectable surface levels of HER2 were reported in the healthy primary cell lines.

We observed that only HA HER2-28Z CAR-T cells (both mock and PD-1 KO) were activated in the presence of healthy cells, while LA HER2-28Z CAR-T cells did not show reactivity of any kind against healthy primary cells. By contrast, both HA and LA HER2-28Z showed similar reactivity in co-culture with a high HER2-expressing control cancer cell line.

These data have been included as **Fig. 6a** and **6b** (CD107- α), **Fig. 6c** (IFN- γ), **Supplementary Fig. 9f** (IL-2) and **Supplementary Fig 8a-e** (results in control cell lines). We have also created a new section in Results entitled ***“LA PD-1 KO display a safer toxicity profile as compared to HA HER2-28Z CAR-T cells”*** to comment on these results: *“Using CAR-T cells resistant to the inhibition by the PD1-PDL1 axis may be an attractive strategy for the treatment of solid tumors. However, safety concerns arise when targeting tumor associated antigens using a high affinity CAR, as it may exhibit poor discrimination between tumor and healthy tissues expressing lower levels of the target antigen. To address this concern, we established co-cultures of LA or HA HER2-28Z CAR-T cells with or without PD-1 KO and a panel of human primary healthy cells including Epidermal Keratinocytes (NHEK), Renal Epithelial Cells (HREpC), Pulmonary Artery Endothelial Cells (HPAEC), Pulmonary Artery Smooth Muscle Cells (HPASMC), all of which have been reported to express low but detectable HER2 densities [20]. Both LA and HA HER2-28Z CAR-T cells demonstrated comparable reactivity against a control cancer cell line expressing high HER2 levels (Supplementary Fig. 8a-8e). However, only HA CAR-T cells were activated in response to co-culture with healthy cells as evidenced by increased production of CD107- α , IFN- γ and IL-2 (Fig. 6a, 6b and 6c, and Supplementary Fig. 8f, respectively), raising safety concerns. Of note, PD-1 KO did not exacerbate the reactivity of LA HER2-28Z CAR-T cells against primary cells from healthy tissues, which showed a toxicity profile similar to non-tumor specific control T cells.”*

The second comment from the reviewer is related to expand on why HA CAR-T cells could pose risk in terms of increased T-cell exhaustion. Our transcriptomic data supports this notion as HA HER2-28Z CAR-T cells express higher levels of genes related to exhaustion such as IRF4, CTLA4, FAS or MAF as compared to LA mock or LA PD-1 KO CAR-T cells. Unfortunately, due to time constraints and the unavailability of preserved serum materials from previous *in vivo* studies to analyze cytokines as suggested by the reviewer, we are unable to provide additional data to further substantiate this observation.

Nevertheless, existing literature also aligns with the notion that high affinity immunoreceptors might be more prone to exhaustion. In the context of TCR-engineered T cells, the use of TCRs with affinities beyond their natural range led to rapid exhaustion and poor persistence *in vivo*, suggesting that increased affinity may adversely affect T cell responses (Schmid, D. A. et al. Evidence for a TCR affinity threshold delimiting maximal CD8 T cell function. *J Immunol* 184, 4936–4946 2010). In the field of CAR-T cells, a study by Caraballo et al. compared CARs targeting GPC3 with different affinities, they demonstrated that low affinity tumor-infiltrating CAR-T cells presented a less exhausted and apoptotic phenotype as their high affinity counterparts (Caraballo Galva LD et al. Novel low-avidity glypican-3 specific CARTs resist exhaustion and mediate durable antitumor effects against HCC. *Hepatology*. 2022 Aug;76(2):330-344. doi:

10.1002/hep.32279. Epub 2021 Dec 27. PMID: 34897774; PMCID: PMC10568540.). This study also observed a greater persistence *in vivo* of low affinity CAR-T cells. In the same line, results from a study focusing on a CD19-targeting CAR with lower affinity than commercial products demonstrated greater persistence in preclinical mouse models and patients in a clinical study (Ghorashian S et al. Enhanced CAR T cell expansion and prolonged persistence in pediatric patients with ALL treated with a low-affinity CD19 CAR. Nat Med. 2019 Sep;25(9):1408-1414. doi: 10.1038/s41591-019-0549-5. Epub 2019 Sep 2. PMID: 31477906.) Collectively, these studies emphasize the need to reconsider the criteria for determining optimal CAR affinities to achieve enhanced therapeutic indices.

We have now modified the manuscript to include the above-mentioned references and the discussion to include the following statements commenting on this issue: *“Our transcriptomic data supports the notion of that HA CAR-T cells may be more prone to exhaustion. In the literature, a recent study demonstrated less exhausted and apoptotic phenotype and greater persistence of CAR-T cells targeting GPC3 with low affinity as compared to their high affinity counterparts. In the same line, a CAR targeting CD19 with lower affinity than commercial products demonstrated greater persistence in preclinical mouse models and patients in a clinical study. Regarding safety concerns, a serious event occurred in the context of HER2-targeting CAR therapy where the use of a HA CAR (based on the scFv 4D5, as employed in our study) led to a fatal outcome in a patient with colon cancer metastatic to the lungs and liver. This was attributed to the high doses of CAR-T cells administered and to the potential CAR-mediated recognition of low levels of HER2 on lung epithelial cells. Our findings evidence a more favorable toxicity profile of LA PD-1 KO as compared to HA HER2 CAR-T cells. This underscores the necessity for caution and thorough investigation when employing HA CARs, emphasizing their potential for unintended activation in the presence of healthy cells expressing lower levels of the target antigen. Interestingly, in terms of efficacy, an analysis of available data from solid tumor CAR-T trials correlating clinical responses to CAR affinity concluded that the use of CARs targeting their antigens with moderate affinity led to best clinical responses as compared to high affinity CARs.”*

I would appreciate if the authors could expand their setting in the lipid bilayer system to take into account the effect of varying target antigen density.

We have now conducted new studies in the lipid bilayer system to include varying target antigen density. Please, see response to comment 4 by reviewer #2 above.

In Figure 5 did the authors expect such a low percentage of polyfunctional cells (less than 5%) in LA CAR-T cell products even when combined with PD1 ablation? It seems very low even compared to the 40% of polyfunctional cells present in HA CAR-T cell products, which again raises the question on the comparison of LA CAR+ PD1 ablation vs HA CAR.

We understand the reviewer’s concern of that LA HER2-28Z CAR-T cells display a lower percentage of polyfunctional cells as compared to HA CAR-T cells in the single-cell secretome study shown in **Fig. 5a-f**. However, we need to acknowledge the fact that, due to technical limitations, experiments with LA and HA CAR-T cells were performed at different times, making direct comparisons challenging. Moreover, in the HA polyfunctional experiment, some technical aspects had been optimized as compared to the initial experiment with LA CARs, likely contributing to observed differences in polyfunctionality. In support of this statement, the intracellular cytokine staining (ICS) assays with LA and HA HER2-28Z CAR-T cells were conducted in parallel, demonstrating comparable frequencies of IFN- γ ⁺TNF- α ⁺ T cells in the mock conditions. In fact, T cells from the same healthy donors used for polyfunctionality studies were also included in the ICS experiments.

We have included the following statement in the original manuscript to highlight the latter fact: *“Of note, frequencies of IFN- γ +TNF- α + T cells in mock groups from LA and HA CAR-T cells were comparable”*.

I would recommend moving to main figure all data concerning co-treatment with anti PDL-1 blocking antibodies, since it brings further support to the authors claim, and expanding on the advantages of a co-treatment vs genetic ablation in discussion.

As suggested by the reviewer, we have now moved all data relative to co-treatment with PD-1/PD-L1 blocking antibodies to main figures.

In particular, we have moved the *in vivo* study previously shown as **Supplementary Fig. 3a** to main **Fig. 2b**. We also included previously not shown *in vitro* data regarding HER2-41BBZ featuring co-treatment with anti-PD1/PDL1 blocking antibodies alongside PD-1 KO results as **Fig. 8c** and replaced **Fig. 8e** by **Supplementary Fig. 7e**, depicting the effects of combining PD-1/PD-L1 blocking antibodies with M11-28Z CAR-T cells *in vitro*.

We also expanded on the advantages of a co-treatment versus genetic ablation in the discussion section by incorporating the following statements: *“Combining CAR-T cells with immune checkpoint antibodies offers other advantages such as more precise and flexible dosing regimen, eliminates the need for further genetic modifications on T cells, and can impact both endogenous T cells and CAR-T cells. In fact, combination of CAR-T cells with PD-1 blocking antibodies has also been explored in clinical trials. This broadens the scope of therapeutic possibilities, emphasizing the adaptability of our findings to diverse PD-1 disruption approaches in the pursuit of enhanced CAR-T cell therapy.”*

Minor points

Did the authors expect and have an explanation as to why in Figure 2 PD1-KO CAR-T cells are less efficient than mock cells in controlling the growth of PDL1-KO tumors?

Acknowledging the reviewer's concern regarding the seemingly reduced efficiency of PD-1 KO LA HER2-28Z compared to mock CAR-T cells in controlling the growth of PD-L1 KO SKOV3 tumors, as illustrated in **Fig. 2a**, it is important to note that although there is a trend, the differences do not reach statistical significance. Consequently, it remains uncertain whether these variations are biologically significant. Moreover, the *in vitro* results indicate a comparable activity between mock and PD-1 KO CAR-T cells (**Fig. 1g**).

One plausible explanation for this observed trend could be that in our mock CAR-T cells, Cas9 or an sgRNA for a safe harbor site is not included during electroporation. Consequently, no double-strand cut is generated in these cells, in contrast to PD-1 KO CAR-T cells, which might confer an advantage to mock CAR-T cells. Nevertheless, it is important to note that further experimentation would be required to substantiate this hypothesis.

In Fig S1b Capan2 cell line seems bi-modal rather than low-expressing.

We agree with the reviewer and have now categorized PD-L1 expression by Capan2 cell line as bi-modal in **Supplementary Fig. 1b**.

Reviewers' Comments:

Reviewer #1:

Remarks to the Author:

The authors addressed all my comments. I still believe that the PCA should be included for transparency, even if it is not entirely conclusive. Perhaps, PC3 might be help. However, I defer this question to the editor.

Reviewer #2:

Remarks to the Author:

The authors have sufficiently addressed all of my earlier comments.

Reviewer #3:

Remarks to the Author:

The paper highly benefitted from the revision process. The authors really committed themselves to respond to all raised issues, both from myself and other reviewers. They have included additional in vitro experiments with another CAR specificity (FR β) with low/high target affinity, cocultures with HER2 expressing healthy cells to assess safety profile, an additional experiment on supported lipid bilayers titrating antigen density and a different cell line (MDA-MB-48) with varying levels of target expression.

I endorse the publication of the manuscript. I have just minor comments listed below.

- Due to the relevance of antigen density to the new version of the manuscript I would suggest including panels presenting HER-2 expression levels both in SKOV3 and HCC1954.
- In line with the point raised by reviewer 2, I would add CAR expression levels (MFI) in Figure 7h-j.
- In the paragraph "Advantages of PD-1KO do not apply uniformly across different CAR constructs", the authors state: "This was also confirmed using CAR-T cells targeting mesothelin with LA (Suppl. Fig. 10d)". However, in figure 8a (and suppl.fig10c), they do not present LA vs HA mesothelin constructs, but mesothelin BBs vs 28s constructs.
- Suppl figure 10d is missing.

REVIEWERS' COMMENTS

Reviewer #1 (Remarks to the Author):

The authors addressed all my comments. I still believe that the PCA should be included for transparency, even if it is not entirely conclusive. Perhaps, PC3 might be help. However, I defer this question to the editor.

We want to thank the reviewer for their positive review of our manuscript. Your comments and constructive feedback have been invaluable in shaping the final version of our work.

Reviewer #2 (Remarks to the Author):

The authors have sufficiently addressed all of my earlier comments.

We want to extend our gratitude for your review of our manuscript. Your insights have truly enriched our work, and we are thankful for your time and expertise.

Reviewer #3 (Remarks to the Author):

The paper highly benefitted from the revision process. The authors really committed themselves to respond to all raised issues, both from myself and other reviewers. They have included additional in vitro experiments with another CAR specificity (FR β) with low/high target affinity, cocultures with HER2 expressing healthy cells to assess safety profile, an additional experiment on supported lipid bilayers titrating antigen density and a different cell line (MDA-MB-48) with varying levels of target expression.

Thank you for your thorough evaluation of our revised manuscript. We are pleased to hear that you found the paper to have benefited from the revision process and that we have addressed the raised issues comprehensively.

I endorse the publication of the manuscript. I have just minor comments listed below.

We appreciate your endorsement of the publication of the manuscript and value your minor comments, which we have now addressed.

- Due to the relevance of antigen density to the new version of the manuscript I would suggest including panels presenting HER-2 expression levels both in SKOV3 and HCC1954.

We agree with the reviewer that showing antigen levels in the tumor cell lines used would be relevant. We have now included panels presenting HER2 expression levels in both SKOV3 (Supplementary Figure 1a) and HCC1954 (Supplementary Figure 1d), as suggested by the reviewer.

- In line with the point raised by reviewer 2, I would add CAR expression levels (MFI) in Figure 7h-j.

We appreciate the suggestion made by reviewer 2 regarding the inclusion of CAR expression levels (MFI) in Figure 7h-j. However, we would like to clarify that the CAR-T cell expansions and CAR staining for the various healthy donors presented in Figure 7h-j were conducted at different time points. As a result, a direct comparison of MFI between these different CAR-T cell products may not be feasible or accurate. Therefore, we believe that adding this data would not provide meaningful insights into the study. Thank you for your understanding.

- In the paragraph "Advantages of PD-1KO do not apply uniformly across different CAR constructs", the authors state: "This was also confirmed using CAR-T cells targeting mesothelin with LA (Suppl. Fig. 10d)". However, in figure 8a (and suppl.fig10c), they do not present LA vs HA mesothelin constructs, but mesothelin BBs vs 28s constructs.

We have revised the wording in this entire paragraph to provide clearer explanations and to avoid confusion. We have not tested CAR constructs targeting mesothelin with high affinity in this study. The CAR targeting mesothelin with low affinity was utilized to (a) validate the sensitivity of a low-affinity CAR to the PD-1/PD-L1 axis and (b) confirm the resistance of 4-1BB-based LA CARs to PD-L1-mediated inhibition.

- Suppl figure 10d is missing.

We apologize for the absence of Supplementary Figure 10d. We have corrected this in the final version of the manuscript.